# A fully integrated, standalone stretchable device platform with in-sensor adaptive machine learning for rehabilitation

Hongcheng Xu [1], Weihao Zheng[1], Yang Zhang [2], Daqing Zhao[3], Lu Wang[3], Yunlong Zhao[4], Weidong Wang [1] ✉, Yangbo Yuan[1], Ji Zhang [1], Zimin Huo[1], Yuejiao Wang [5], Ningjuan Zhao[1], Yuxin Qin[1], Ke Liu[1], Ruida Xi[1], Gang Chen[1], Haiyan Zhang [1], Chu Tang [6], Junyu Yan[1], Qi Ge [7], Huanyu Cheng [8] ✉, Yang Lu [9] ✉ & Libo Gao [4] ✉

Post-surgical treatments of the human throat often require continuous monitoring of diverse vital and muscle activities. However, wireless, continuous monitoring and analysis of these activities directly from the throat skin have not been developed. Here, we report the design and validation of a fully integrated standalone stretchable device platform that provides wireless measurements and machine learning-based analysis of diverse vibrations and muscle electrical activities from the throat. We demonstrate that the modified composite hydrogel with low contact impedance and reduced adhesion provides high-quality long-term monitoring of local muscle electrical signals. We show that the integrated triaxial broad-band accelerometer also measures large body movements and subtle physiological activities/vibrations. We find that the combined data processed by a 2D-like sequential feature extractor with fully connected neurons facilitates the classification of various motion/speech features at a high accuracy of over 90%, which adapts to the data with noise from motion artifacts or the data from new human subjects. The resulting standalone stretchable device with wireless monitoring and machine learning-based processing capabilities paves the way to design and apply wearable skin-interfaced systems for the remote monitoring and treatment evaluation of various diseases.

Wearable devices start to gain momentum in disease diagnostic confirmation, treatment evaluation, and healthy aging[1–4]. For those who suffer from congenital choking[5,6] or neck cancer[7–9], post-surgical rehabilitation of the human throat often requires the clinician's continuous monitoring and evaluation of swallowing ability[10], vocal-fold motion[11], oral intake of liquid[12], and others[13–15]. Monitoring and diagnosis of these behaviors can also prevent secondary injuries that often occur in patients with dysphagia disorders during normal daily activities. Currently, the commercial devices to track laryngeal signatures are rigid, bulky, and tethered, including the lip-closing force gauge[16],

non-invasive belts[17–19], thin-film pressure sensors[20], and others[21]. The poor device/skin contact also results in attenuated signals that are also susceptible to motion artifacts. Therefore, soft on-throat devices are urgently needed to continuously monitor laryngeal activities for diagnosis and rehabilitation evaluation.

Diverse laryngeal activities with serial muscle movements typically cause laryngeal muscle motions and local inertial vibrations[22]. Therefore, they are often used to evaluate the laryngeal health condition of patients. For instance, swallowing amplitude and rhythm reflect the intake ability of food and water[23]. However, features

---

obtained by traditional wearable devices such as the single force sensor on the throat[20] can only provide limited information about the patient's health condition. Monitoring of both vibrations and muscle activities is mainly achieved by using separate inertial devices and force sensors on the skin. Efforts to address this challenge lead to the development of a wearable accelerometer for neck voice disorders[24] and flexible surface electromyogram (sEMG) electrodes with a strain sensor for oropharyngeal swallowing disorders[23,25]. However, these flexible sensors suffer from limited stretchability of less than 16%[23,25], poor system integration with only sensors without functional circuit board[21,26], and severe skin inflammation or allergy during use over several hours owing to its low permeability[27–29]. It is important to integrate soft wearable electronics[2,30,31] with data processing/transmission units[32–34] to realize the full potential of the wide range of electro-mechanical signatures[35,36]. A recent development on a skin-mounted mechano-acoustic sensing system provides a prospective way to track the activity at the suprasternal notch[37,38]. However, high-quality sEMG signals without being affected by motion artifacts are yet to be integrated into the device platform and the data analysis based on the advanced machine learning algorithm on the cloud is still needed for remote monitoring and evaluation.

Machine learning-based diagnosis is of great interest in the development of smart medicine, especially integrated with soft electronics (to classify the dysphagia severity)[37,39,40]. Early efforts include the use of the one-dimensional (1D) convolution neural network (CNN)-based deep learning for rehabilitation monitoring after orthopedic surgery[41] and predicting knee joint postures[42]. Despite high recognition accuracy, these CNN models with individual-1D data sources only offer limited feature information during the learning processes for target predictions. These models without memory function and adaptive capabilities suffer from low prediction accuracies for data from new subjects, which is critical in practical applications.

Herein, this work presents a fully integrated standalone stretchable device platform that can wirelessly measure and analyze diverse vibrations and muscle activities directly from the human skin. The modified composite hydrogel electrode interface is designed to maintain robust contact to the throat with low contact impedance for improved signal quality during motion and low adhesion for easy removal. Besides sEMG signals, the triaxial broad-bandwidth accelerometer integrated into the patch can also monitor large body movements (e.g., walking and jumping) and subtle physiological activities (e.g., heartbeats and respiration). With a 2D class sequence feature extractor based on the CNN algorithm, 13 general features from fourteen healthy human subjects and two patients (one with myasthenia gravis and the other with laryngeal cancer) can be classified with a high accuracy of 98.2%. More importantly, the fully connected neurons of the 2D-like sequential feature-extracting model can allow the device system to adapt for use with noise from motion artifacts and on new subjects with a high prediction accuracy of 92%. A wireless user interface further enables remote monitoring and real-time evaluation of laryngeal activities on the cloud server, paving the way for the next-generation standalone stretchable device platform for laryngeal rehabilitation management and diagnosis and treatment evaluation of various diseases.

## Results and discussion
### Design of the laryngeal patch
The standalone stretchable device platform consisting of hydrogel electrodes and functional electronic components with signal-processing units interconnected by the coplanar serpentine Cu network (Fig. 1a) can directly adhere to the human skin (Fig. 1b). The processed and wireless transmitted signals from the inertial triaxial accelerometer and hydrogel electrodes (Fig. 1c and Fig. S1) provide continuous and non-invasive monitoring of local vibrations and

muscle activities from the larynx and other locations on the human body. Various activities can also be distinguished with an efficient convolutional neural network and the data processed on a cloud server further facilitates remote rehabilitation and disease diagnosis (Fig. 1d). The standalone stretchable device platform fabricated from low-cost processes (Figs. S2−5) exhibits robust electromechanical performance upon various mechanical deformations (e.g., stretching, bending, and twisting) as verified by both finite element analysis (FEA) and experiments (Figs. 1e, S6, and movies S1−3).

### Design and characterization of the composite hydrogel interface
The composite hydrogel mainly consists of monomer ([2-(Methacryloyloxy)ethyl]dimethyl-(3-sulfopropyl) ammonium hydroxide, DMAPS), crossed linker, photo-initiator, and ionic salt (see fabrication in Method). Without a need to strictly balance positive and negative charges during the copolymerization process, a wide range of monomer ratios, concentrations, and ionic strengths can be used to synthesize in this zwitterionic-type hydrogel[43]. To avoid oxidation of the copper-based electrode array and improve the contact quality to the skin for enhanced signal acquisition, a stretchable and highly conductive ionic hydrogel interface is designed by doping Ag nanowires (AgNWs) in the ionic composite with a polydimethylsiloxane (PDMS) skeleton (Fig. 2a). The design with the PDMS skeleton exhibits a reduced peak strain of 0.86 for a uniaxial stretching of 50%, compared to that of 3.13 in the one without (Fig. 2b). The stretchable PDMS with a higher modulus (~200 kPa) and optimized number of vertical beams (Fig. S7) provides the modified hydrogel composite with improved load and strain bearing capabilities (14 kPa/200% for the tensile stress/strain) (Figs. 2c and S8). The AgNWs with optimized concentration (0.7 wt%, Figs. S9−10) in the hydrogel not only provide high conductivity, but also result in lower contact impedance than that of the commercial gel electrode (Figs. 2d and S11) for the high-quality acquisition of sEMG signals (Fig. 2e). The lower contact impedance results from the high conductivity at the Cu/hydrogel interface (Fig. S12) and the improved hydrogel/skin contact quality as observed on the skin replica (Fig. 2f). The increased contact impedance from drying can recover by applying pure water (Fig. S13). Meanwhile, it is interesting to note that the composite hydrogel presents a lower tensile strain (Fig. 2g) and ca. eight times smaller peeling force (Fig. 2h) compared with the commercial gel electrodes, which facilitates easy removal, especially from the skin of the infants or elderly. The composite hydrogel that can protect the skin from UV and IR radiation (Fig. S14) also exhibits high cell viability (>80% of epithelial cells) and biocompatibility (Fig. 2i) for long-term monitoring. The modified hydrogel with excellent contact impedance and other properties suitable for integration with flexible circuits in the standalone device platform for health monitoring is superior compared to the previously reported hydrogels[27,44−60] (Table S1).

### Electrical and mechanical performance of the fully integrated patch
The fully integrated device is designed to provide high-quality wireless transmission. As shown in electromagnetic (EM) simulations, a miniaturized low-temperature co-fired ceramic (LTCC) antenna (Fig. S15) is integrated with the low-power Bluetooth module to investigate and demonstrate the transmission properties of the device. The LTCC antenna on the Cu serpentine network further encapsulated by a 500 μm-thick Ecoflex layer only shows a small frequency shift (<19 MHz) and maintains excellent impedance matching (with a return loss <10 dB) (Fig. 3a), leading to negligible impact on the near and far radiation fields (Fig. 3b−d). The biocompatible Ecoflex encapsulation also isolates the thermal radiation (Fig. 3e) even when the patch is powered by a lithium-ion battery (Fig. S16) due to low power consumption, which is suitable for long-term use on the skin. The tight coupling of the device to the skin provides high-quality sEMG signal

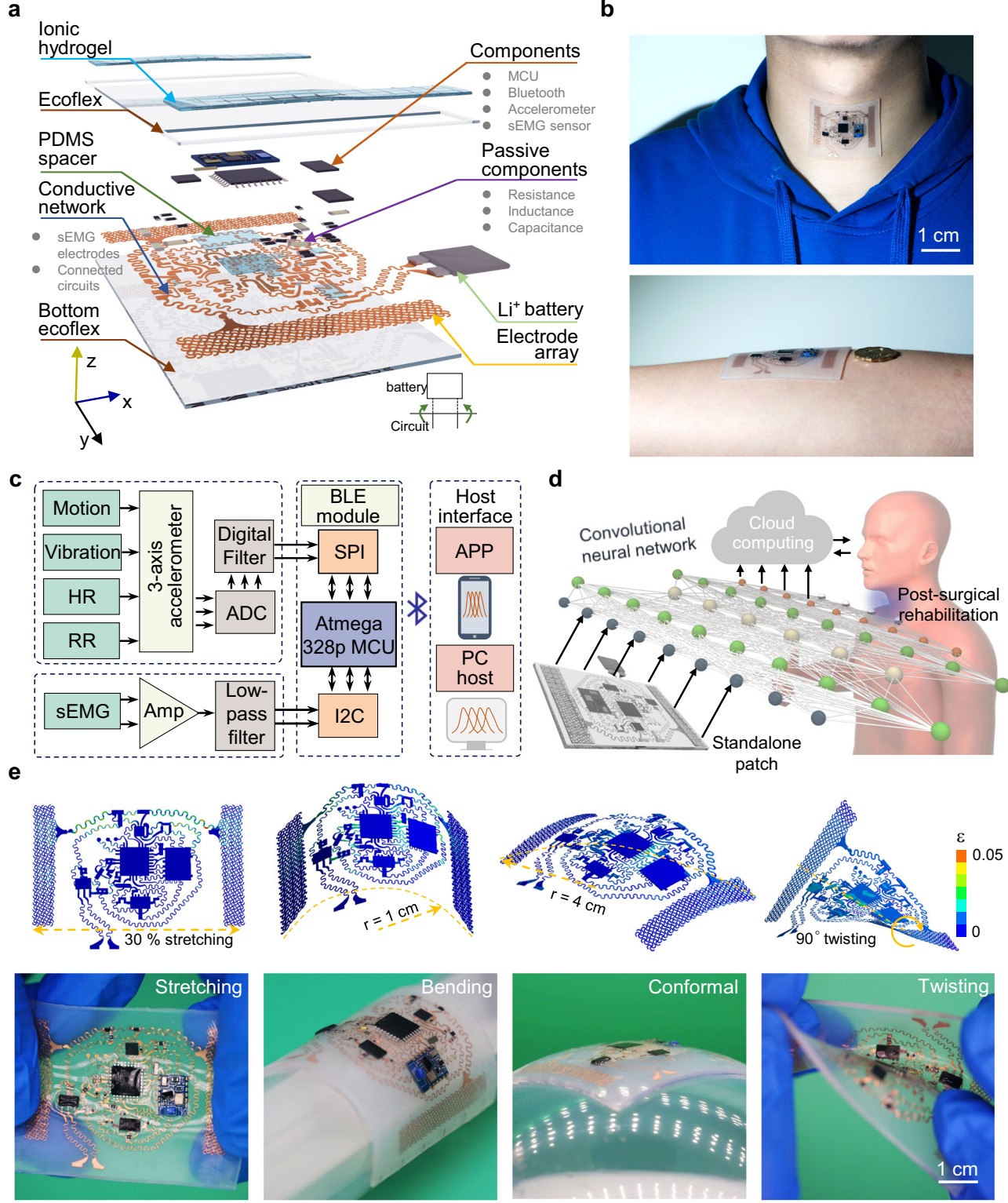

**Fig. 1 | Overview and design of the standalone stretchable laryngeal patch.**
**a** Exploded diagram of the integrated device system. **b** Optical images of the standalone stretchable device patch attached to the laryngeal skin (top) and forearm (bottom). **c** Block functional diagram showing the processing steps of the acceleration and surface electromyography (sEMG) signals, including signal processing, controlling, communication, and display. **d** Schematic showing the use of a machine learning network and the standalone stretchable patch in laryngeal post-surgical rehabilitation. **e** Finite element analysis (FEA) and corresponding experimental results of the patch under mechanical deformations: uniaxial stretching of 30%, bending to the cylinder with a radius of 1 cm, conforming to a sphere with a radius of 4 cm, and twisting with a torsional angle of 90°.

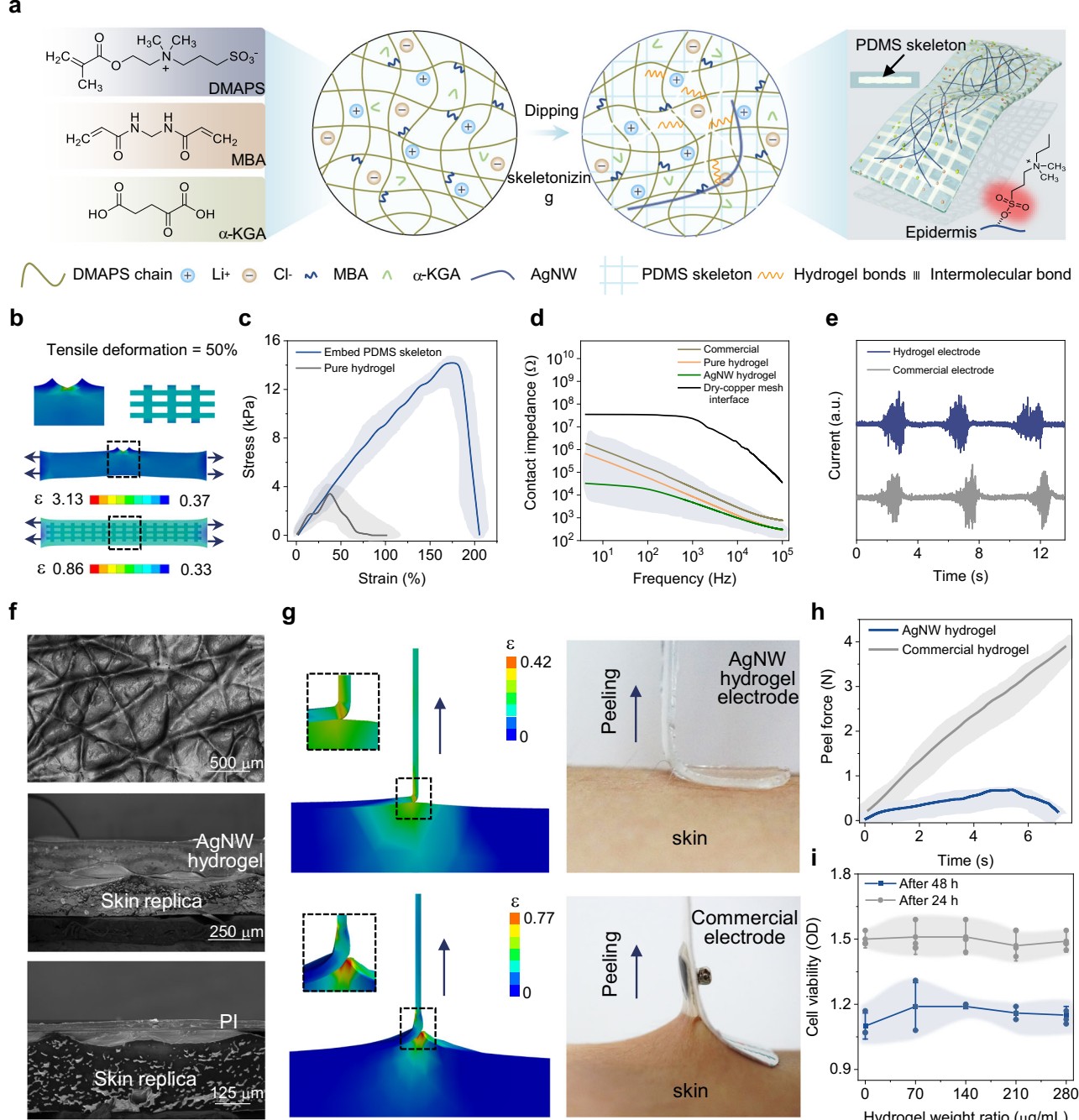

**Fig. 2 | Characterizations of composite hydrogel electrodes. a** Schematic showing the composite hydrogel with PDMS skeleton. **b** Comparison of the strain distribution between the hydrogels with and without PDMS skeleton under 50% uniaxial stretching. **c** Stress-strain curves of the hydrogels with and without PDMS skeleton under uniaxial tension. **d** Comparison in the conductivity of electrodes based on the commercial gel, copper mesh, pure hydrogel, and hydrogel mixed with AgNW. **e** Comparison of sEMG signals acquired by composite hydrogel and commercial gel electrodes. The term (a.u.) in (**e**) represents the arbitrary unit. **f** Scanning electron microscope (SEM) images of the skin replica (top), the composite hydrogel (middle), and the polyimide (PI) (bottom) on the skin replica. **g** Simulated strain distributions (left) and optical images (right) of the composite hydrogel and commercial gel electrodes on the skin during vertical peeling. **h** Measured peeling forces of the hydrogel and commercial gel electrodes from the forearm. **i** Biocompatibility tests of epithelial cells cultured in the uncured hydrogel solution. Error bars in (**i**) represent standard deviations, $n = 3$ independent samples. Data in (**i**) are presented as mean values ±SEM. Error bands in (**c, d,** and **h**) represent standard deviations, $n = 3$ independent samples.

even with motion artifacts such as from "talking", as evidenced by the highest power spectral density and the lowest signal-to-noise ratio compared to commercial devices (Figs. 3f, g and S17).

The device also exhibits high mechanical compliance and robustness, including a skin-like Young's modulus of 89.5 kPa (Fig. S18) and a small force hysteresis of <6.4% for a tensile strain of 30% (Fig. 3h), leading to comfort integration on the skin upon various deformations

(Fig. 3i). Even when the device is stretched to 30% that is the maximum strain on the skin[61], the maximum principal strain (3%) in the Cu serpentine network (Fig. 3j) is still below the fracture strain of Cu (5%)[62]. The serpentine design of the Cu network also helps the device patch to maintain the mechanical properties such as the stress-strain curve (Figs. S19–20) for high-quality monitoring during skin/body movements. The fully integrated device with superior mechanical and

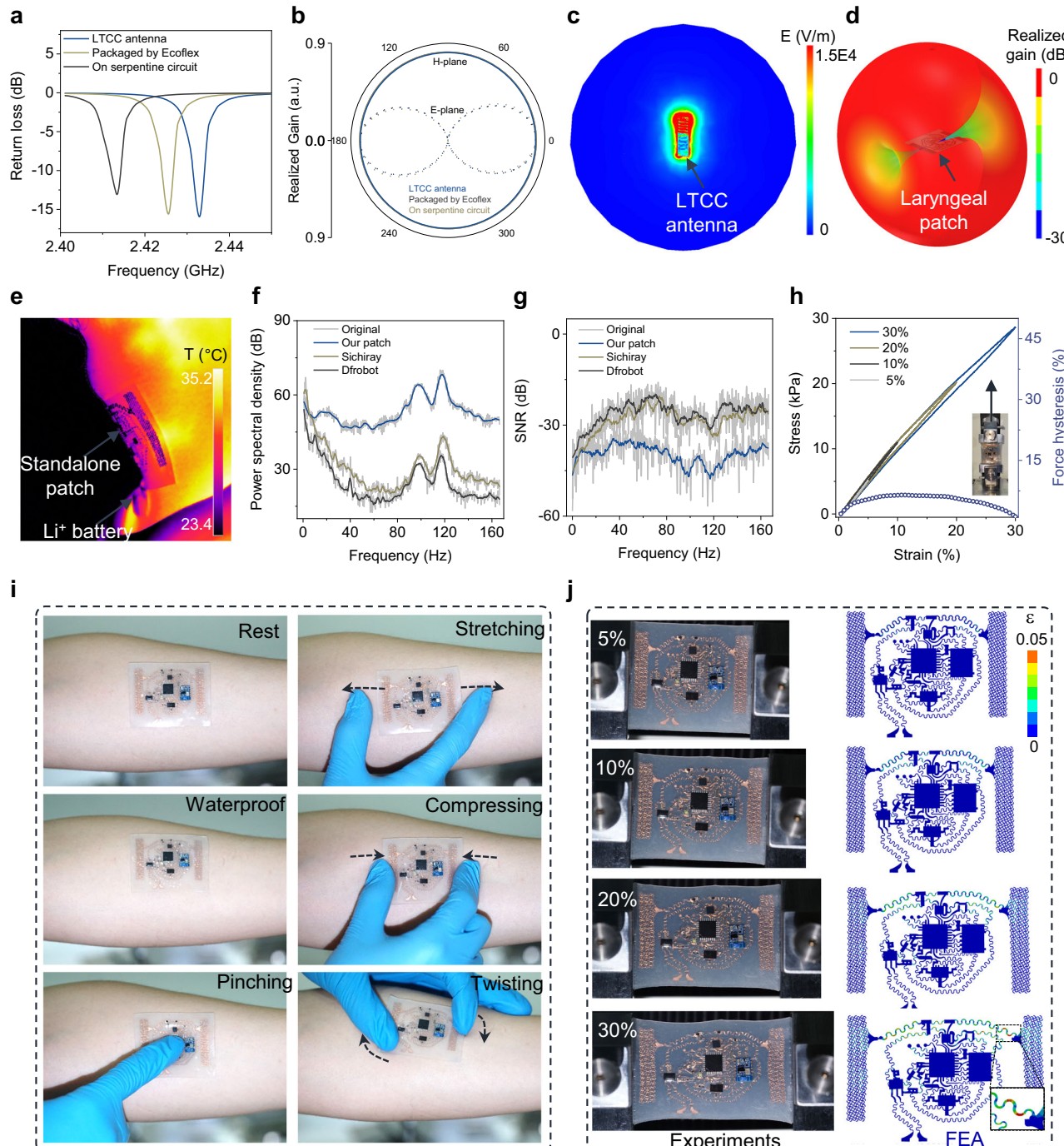

**Fig. 3 | Electronical and mechanical characterizations of the patch.** Comparisons of (**a**) return loss and (**b**) radiation pattern between the low-temperature co-fired ceramic (LTCC) antenna before and after packaged by Ecoflex and integrated with the stretchable serpentine circuit. The term (a.u.) in (**b**) represents the arbitrary unit. **c** Electrical field pattern and (**d**) far radiated field of the fully packaged/integrated LTCC antenna in the patch at the resonant frequency. **e** Heat map as the patch operated on the laryngeal skin of a volunteer. Comparisons of (**f**) power spectral density and (**g**) signal-to-noise ratio (SNR) of the sEMG signals by the Welch transform measured from our patch and commercial devices (i.e., Sichiray and Dfrobot) on the laryngeal skin. **h** Force hysteresis of the patch under uniaxial stretching from 5% to 30%. **i** Optical images of the patch on the forearm under rest, watering, pinching, stretching, compressing, and twisting conditions (no external adhesive). **j** Comparison between the experiment and FEA results of the patch under a uniaxial tensile strain up to 30%.

electrical properties for conformable monitoring (e.g., swallowing activities) during movements outperforms previous reports of laryngeal sensors[16,23,24,38,63–76] (Table S2).

## Monitoring of various biophysical activities at the throat
As the throat provides rich information on vibrations and muscle activities for clinical diagnosis of various diseases and corresponding post-surgical training/evaluation[15], the device is applied to this critical location for a proof-of-conception demonstration. The high-sensitivity accelerometer ADXL-345 with a sampling frequency of up to 800 Hz in the patch allows successful continuous monitoring of various activities, such as sitting, talking, swallowing, walking, and jumping (Fig. 4a and movies S4–6), with a wide frequency spectrum from 0 to 400 Hz (Fig. 4b). The simultaneously measured acceleration data along three different directions allow the integrated system to distinguish multiple motions separately (e.g., talking while walking, drinking water while

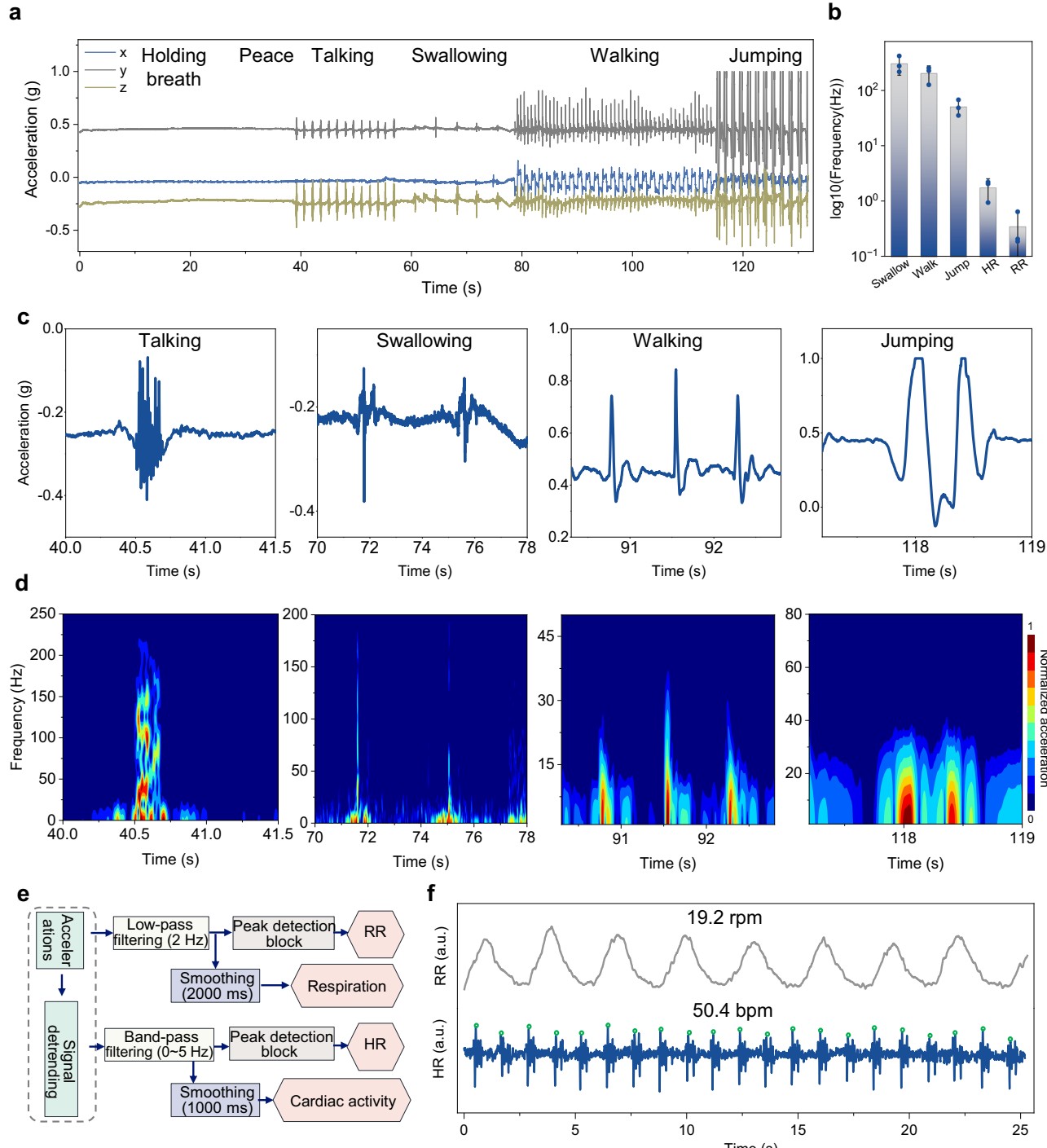

**Fig. 4 | Wireless recording of physiological processes and body motions.**
**a** Long-term continuous recording of various activities (e.g., sitting still, talking, swallowing, walking, and jumping). **b** Frequency spectrum of various biophysical activities of a healthy human. Error bars represent standard deviations, *n* = 3 independent samples. Data in (**b**) are presented as mean values ±SEM.

Representative data (**c** and **d**) their corresponding frequency performance via the short-time Fourier transform (STFT). **e** Schematic block diagram of the algorithm and (**f**) the decoupled heart rate (HR) and respiration rate (RR) from inertial acceleration signals in real-time. The term (a.u.) in (**f**) represents the arbitrary unit.

coughing, drinking water while swallowing) (Figs. S21 and S22), as well as capture the swallowing process from a patient with myasthenia gravis (Fig. S23). In the laryngeal events, the talking and swallowing signals follow the vibrational movements along the normal direction of the skin (z-axis), whereas walking and jumping signals stem from body motions along the throat skin from the neck to the head direction (y-axis). The time-frequency contour of the signals from the short-time Fourier transform confirms the high-frequency (up to 200 Hz)

responses with a lower amplitude (less than 0.4 g) of swallowing and talking (Fig. 4c, d). In contrast, walking and jumping are opposite with the slow-moving frequency and a large acceleration magnitude (Fig. S24). These features are highly consistent with the behavior of adults[38]. In addition, periodically weak vibrations of laryngeal bones from cardiac and lung dilatation processes provide routes to measure heart rate (HR) and respiration rate (RR) from recorded acceleration along the Z and Y axes, respectively. With digital filtering and peak-detection

(Fig. 4e), the HR and RR of 19.2 and 50.4 times per minute can be decoupled from the original data (Fig. 4f). As the mounting position of the patch moves up the laryngeal skin, the measured cardiac behavior remains unchanged, but the respiration amplitude decreases (Fig. S25) due to significantly reduced movements (along the neck direction) farther away from the chest cavity. Moreover, the swallowing signature can be clearly observed by placing the integrated device at the throat area with comparable performance to the suprahyoid area, whereas the signal at the suprasternal notch area is almost inconspicuous (Fig. S26), leading to the choice of middle throat area for laryngeal detection. More importantly, the combination of the acceleration data with sEMG for muscular activities during talking (Fig. S27) can further decouple speaking for speech recognition and more precise diagnosis and evaluation of various diseases in the clinical practice.

## The machine learning model for rehabilitative evaluation

The clinical evaluation for laryngeal rehabilitation usually focuses on typical events (i.e., talking, swallowing, volume, and viscosity swallowing with modified safety and effectiveness indicators) with long-term and extensive efforts[77]. To help automatically evaluate the laryngeal condition of the new patients and healthy individuals, a CNN-based 2D-like sequential feature extractor (2D-SFE) is explored to classify and infer pathological status based on the classification of physiological events (Fig. 5a). Comparison in the overall accuracy between this work and previous reports indicates the advantage of integrating two distinct signal inputs in adaptive machine learning[34,71,73,76,78–82] (Table S3). The collected 1D data (acceleration and sEMG) are first transferred to a 2D vector similar to an image matrix for processing by the CNN-based 2D-SFE that contains 62 filtering layers (Fig. S28) and 2 classifying layers (Fig. S29). In the training model with the above-related activities, the processing vectors are iterated from the convolutional layer to the pooling layer and then to the activation layer to achieve dimensionality reduction of the feature vectors (FVs). The extracted feature is further classified into special targets by evaluating the consistency of the softmax function in the full connection layer. In the proof-of-the-concept demonstration, five Chinese pinyin and five vowels are chosen as the acoustic states, together with swallowing, drinking water, and coughing behaviors as feature states from fourteen healthy human subjects and two patients (one with myasthenia gravis and the other with laryngeal cancer, sampling rate of 333 Hz). The acceleration and sEMG data in 2D-like vector contours (Fig. 5b and SI Note1) also include noise from other motion actions (e.g., drinking water while coughing) to mimic real-world situations. Randomly dividing each 2D data of the laryngeal feature into 100 sequences with a fixed length of 1000 data points ensures data reliability (covering the entire test data of each feature). The triplet and cross-entropy are used as the objective functions for feature extraction and classification, respectively (Fig. 5c). Finally, the classified feature ultimately corresponds to the tested states, forming the confusion matrix[83].

After training these features through 62 extracting layers and 2 classification layers, the accuracy of all states from training and testing data tends to peak at 60 epochs (Fig. 5d), which can also be verified by the normalized loss for the two objective functions during 100 iterations (Fig. 5e). FVs clustered by the t-distributed stochastic neighborhood embedding algorithm during the training process help visualize the evolution of the principle component in the machine learning space (Fig. 5f). The machine learning model achieves an overall prediction accuracy of 98.2% for the 13 states/features studied in this work (Figs. 5g and S30), demonstrating the excellent performance of the CNN-based 2D-SFE for multidimensional vector prediction (e.g., multiple sequential targets over time)[78].

## Applications for remote diagnosis and monitoring

The measured data wirelessly transmitted to the cloud server in real-time via the local cellular network also creates opportunities for remote diagnosis and treatment evaluation over time (Fig. 6a and S31). In the proof-of-the-concept demonstration, the free Ali-cloud is chosen as the cloud hub and the user interface is custom-built according to the application development protocols. Validation of the CNN-based 2D-SFE with the laryngeal activities from another two subjects (SI Note1) shows a high accuracy of over 95% for the 13 features along with three performing states to simulate movement artifacts from choking, chewing, and nodding (Fig. 6b). To help evaluate the post-surgical state, a pathological rehabilitative degree is sorted into eight levels in the cloud-served interface based on three typical activities: swallowing, talking, and drinking water (*Methods*). The machine learning model with fully connected neurons can adapt to events from new human subjects, resulting in an overall classification accuracy of up to 92% (Fig. 6c). The challenge to separate the principal components of the three typical activities from talking (Fig. 6d) accounts for the relatively low prediction accuracy (Fig. S32), but the overall prediction accuracy still reaches 92%. Reducing the feature category from 13 to 4 (still from 13 different behaviors) leads to quick convergence of the adaptive system (Figs. 6e, f and S33) with a high accuracy of 89.7% (see *Methods*), demonstrating its power for use in new human subjects.

The decoupled measurements of HR, RR, and various behaviors over a long term are critical for precise diagnosis and treatment evaluation. Besides the good agreement in the HR between our patch and a commercial platform (BL-420) during sitting and breath-holding (Fig. 6g), the characteristic R, S, and T waveforms can also be clearly observed during long-term monitoring (Fig. 6h). Similarly, the RR tracking from our patch also highly correlates with that from the commercial device, an average RR of ca. 20.35 calculated from the peak-seeking algorithm (Fig. 6i). The relatively long-term monitoring required for perioperative or rehabilitation is further demonstrated by tracking various physiological events for 2.65 h (Fig. 6j, top). Compared with the commercial counterpart, our device exhibits a larger amplitude in the time domain due to the high mechano-acoustic signal quality. The distinguished frequency spectrum width and power intensity can also help decouple these signals (Fig. 6j, bottom). The difference in the fundamental frequency (e.g., a higher value of over 100 Hz from talking and coughing compared with that of other events) allows the evaluation of various laryngeal conditions.

The swallowing process with the pear-shaped postcricoid area transitioned from opening to closing can be accurately monitored by our integrated wireless platform, which is also validated against the gold standard based on the fiberoptic endoscopic examination of swallowing (FEES, Figs. S34–34 and Movie S7). Compared with the healthy control (Fig. S34a), the patient with myasthenia gravis shows higher muscle force to complete swallowing, where this difference is not captured by the FEES (Fig. S34b).

The volume and viscosity swallowing test-chinses version with modified safety and effectiveness indicators is used to quantify the analysis as the subject ingests a blue edible indicator with a body weight percentage of 2% (stage #1) and 1% (stage #2) (Fig. S35a). The clinical standard for swallowing rhythm requires swallowing to be less than 3 s from the indicator in mouth to the circumpharyngeal muscle opening and a total time of less than 15 s for the complete ingestion process. The swallowing in both stages for the patient with myasthenia gravis exceeds 15 s (Fig. S35b). Liquid residual still exists after 25 s in the oropharyngeal junction area in stage #1 (Fig. S35c). The duration of the process of about 10 s captured by our device is consistent with the clinical standard (Fig. S35d), demonstrating the feasibility and reliability of the integrated system. It is important to note that one portion of the ingesting process (7–12 s) in the stage #2 is not captured by the FEES due to the closed oropharyngeal junction area blocking the endoscopic view (Fig. S35e). In contrast, this missing swallowing process is still clearly recorded by our device (Fig. S35f, green dashed ellipse), providing enhanced diagnostic accuracy for laryngeal post-operative patients. Above all, the standalone stretchable device

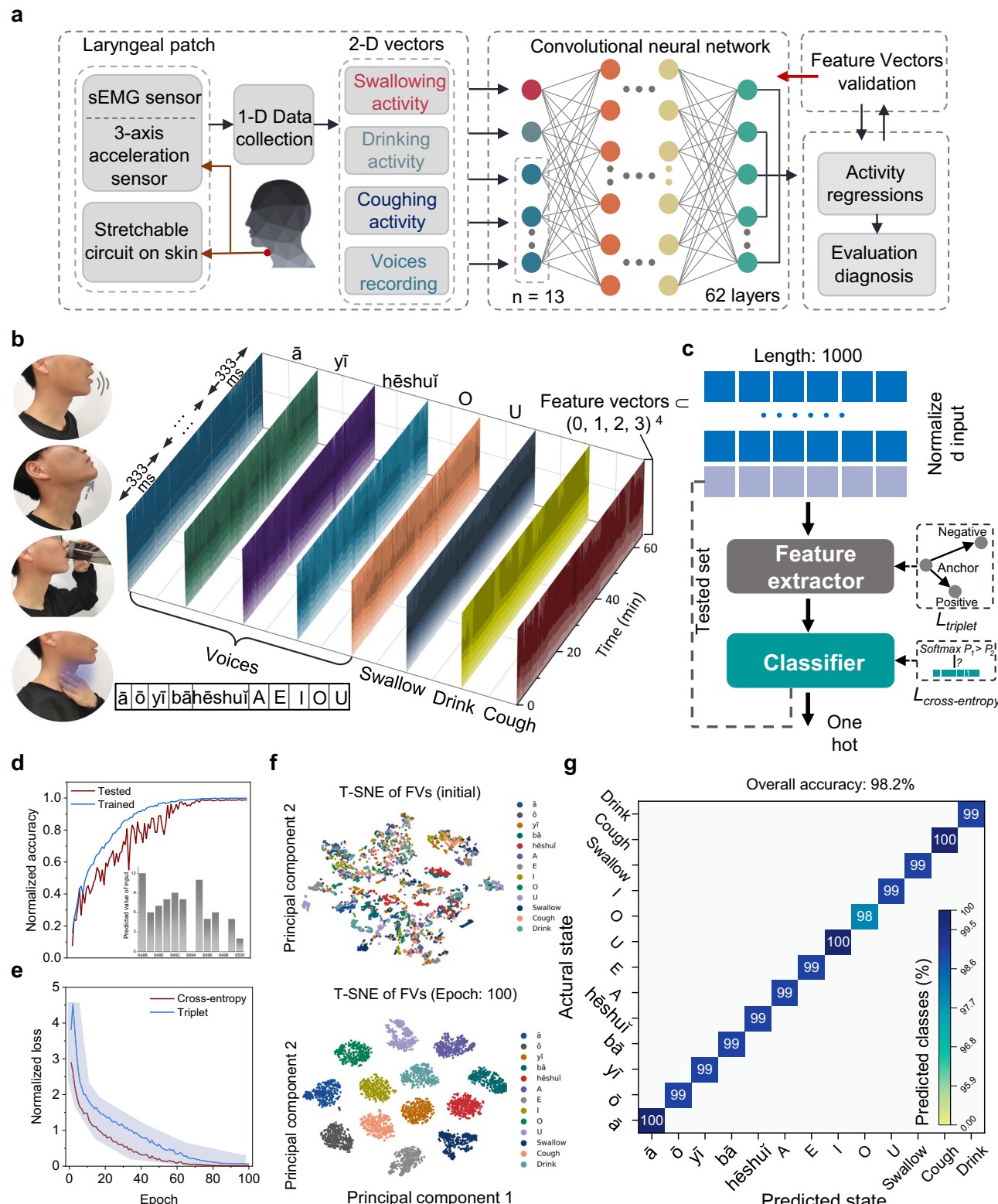

**Fig. 5 | 2D-like sequential feature computing for post-surgical rehabilitation.**
**a** Schematic flow diagram of the machine learning algorithm for training and classifying the swallowing, drinking, coughing, and talking activities. **b** 13-feature contours from the above-mentioned events acquired by the two sensing modalities. **c** Feature vector processing flow of the input signal through machine learning. **d** Normalized accuracy during 100-epoch iterations for the training and testing data. **e** Normalized loss of the triplet and cross-entropy functions to optimize feature extraction and goal classification, respectively. Error bands in (**e**) represent standard deviations, $n = 3$ independent samples. **f** Feature vector matrix before (top) and after 100-epoch iterations (bottom) processed by the t-distributed stochastic neighborhood embedding (T-SNE) algorithm. **g** Confusion matrix of the 13 features after 100-epoch iterations.

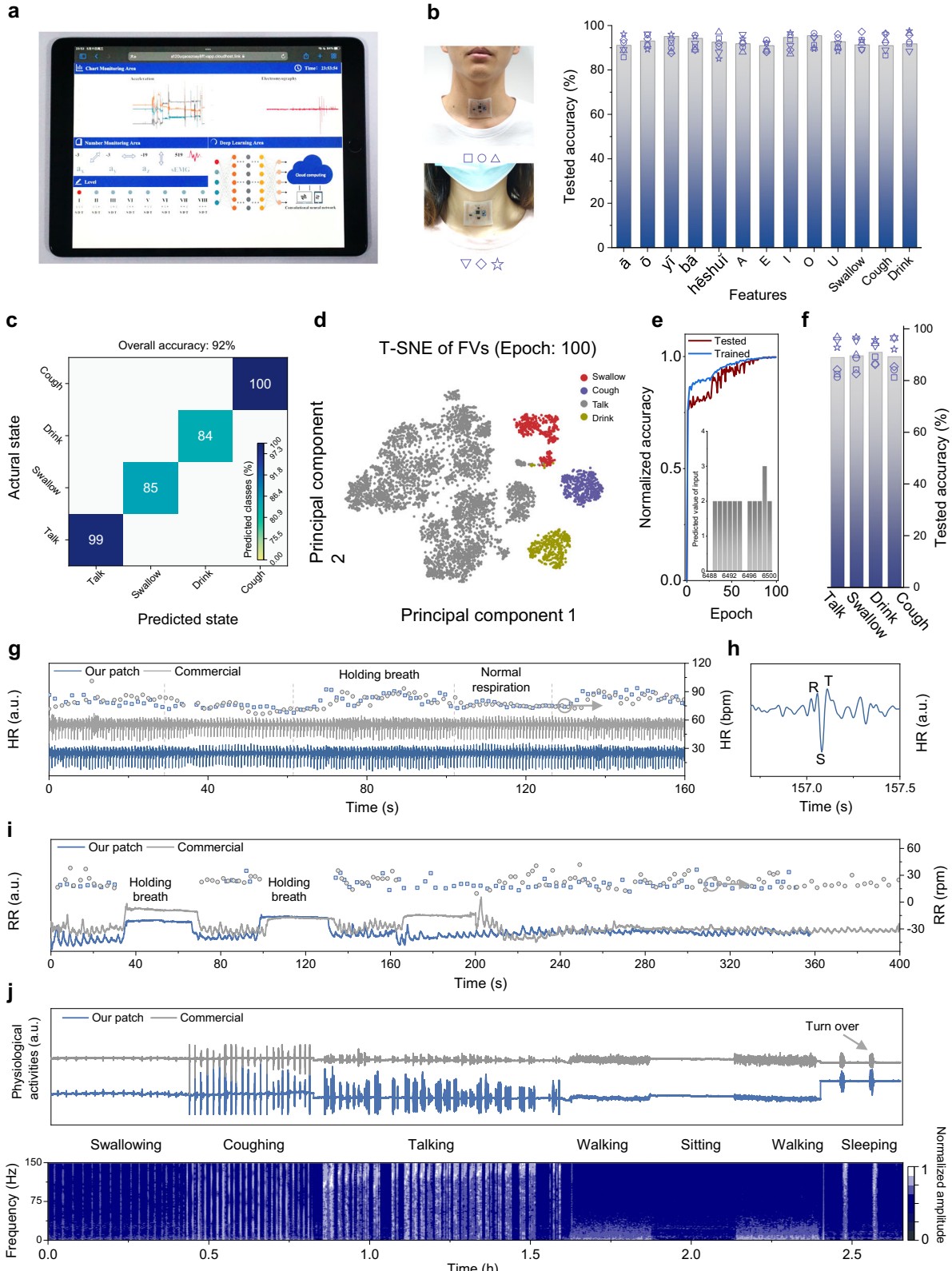

**Fig. 6 | Applications of wireless rehabilitative evaluations. a** An optical image of a cloud-based user interface to monitor inertial acceleration and sEMG signals online for clinical diagnosis. **b** Prediction accuracy of the aforementioned 13 features from two new human subjects. **c** Confusion matrix for four features from the adaptive model. **d** Final feature vector matrix after 100-epoch iterations through T-SNE dimension reduction. **e** Normalized accuracy of the modified machine learning system for four typical activities from the training and testing data, with the final normalized predicted value of the features shown in the inset. **f** Predicted accuracy of four features from the two new human subjects. **g** Comparison of HR acquired using our patch and commercial device on the laryngeal skin of a male volunteer, with (**h**) expected R, S, and T peaks. **i** Comparison of RR between our patch and commercial device. **j** Long-term monitoring of biophysical activities using the laryngeal patch for 2.65 h (top) and its corresponding time-frequency spectrum by STFT (bottom). The term (a.u.) in (**g**–**j**) represents the arbitrary unit.

platform with wirelessly long-term monitored data processed by machine learning on the cloud provides a unique tool for diagnosis and rehabilitative evaluation.

In summary, we report a fully integrated standalone stretchable device platform for wireless monitoring and machine learning-based processing of diverse vibrations and muscle activities directly from the skin. The design of the modified composite hydrogel interface provides a conform contact with the human skin, which achieves lower contact impedance for high signal quality during motions and reduced adhesion for easy removal. The integrated triaxial accelerometer with a broad bandwidth can also accurately monitor both large motions (e.g., walking and jumping) and subtle vibrations (e.g., heartbeats and respiration). The measured data from 13 general signatures/states during laryngeal activities processed by a 2D-like sequential feature extractor based on the CNN can be classified with a high accuracy of 98.2%. The fully connected neuron in the machine learning model further allows it to accurately classify the data from new human subjects with a 92% accuracy. In addition, the custom-built interface to process the data on the cloud opens up new opportunities for remote diagnosis and treatment evaluation for rehabilitation management and various disease applications. Above all, the developed standalone stretchable platform integrates the skin-interfaced soft electrodes and sensors for biophysical monitoring with a stretchable hybrid circuit for wireless transmission. Furthermore, the sEMG combined with 3D acceleration signals captured by our integrated device platform can provide rich information for the feature extractor to adapt to new human subjects, allowing evaluation and rehabilitation of swallowing disorders.

## Methods

### Ethics declaration
All human subject studies were approved by the Institutional Review Board of the First Affiliated Hospital of the Air Force Medical University (protocol: KY20222259-C-1), and the volunteers gave informed consent. The authors affirm that human research participants provided informed consent for publication of the images in Figs. 1b, 5b, and 6b.

### Fabrication of the stretchable patch, hydrogel interface, and LTCC antenna
The fabrication of the stretchable patch primarily comprises (i) the engraving and transfer of the conductive serpentine traces, (ii) the low-temperature reflow soldering of components, and (iii) the encapsulation with Ecoflex (Fig. S2). Firstly, the mixed PDMS precursor with a weight ratio of 10:1 for the base to curing agent (Dow Corning, Sylgard 184A to B) was spin-coated on a clear glass plate at 600 rpm for 10 s, followed by curing at 70 °C for 1.5 h to form the adhesive substrate. Next, a copper (Cu) foil with a thickness of 8 μm (Red copper, T1100), coated on polyimide (PI, 3 μm, HanKe New materials Co., LTD.), was laminated on the PDMS film at a pressure of 100 kPa for 1 h. With the designed CAD file, a 355 nm UV laser (Yuanlu corporation, Wuhan) was used to engrave the serpentine Cu structure with 100 kHz pulse frequency at a speed of 300 mm s$^{-1}$ for 4 times repeated cutting. Peeling off the residual left the conductive Cu network on the PDMS. The serpentine trace was then transferred to the uncured Ecoflex elastomer (Smooth-on, USA) mixed at a weight ratio of 1:1 (A: B) by a water-soluble tape (AQUASOL). Applying deionized water for 3 h dissolved the water-soluble tape. After placing laser-engraved rectangle PDMS isolators at the designed chip location, a thin layer of solder (AL656, Abond) was printed on the connecting pads through a laser-engraved PET mask. After placing all chips and components (smaller than 0.5 cm$^2$, Fig. S36), the entire patch was heated in a solder pot (ZB2520HL, HuaQi zhengbang) at 138 °C for 30 min. Next, the water-soluble tape was applied to the copper mesh electrode, and an insulting oil (PVB, Langyi Chemical, Zhongshan) was sprayed on the circuits (except the water-soluble tape), followed by curing at 50 °C for 3 h. Finally, spin-coating silicone elastomer on the bare circuit at 500 rpm for 10 s and curing at 40 °C for 1.5 h completed the fabrication. The high reproducibility of the fabrication method is highlighted by the batch production of the integrated device platform (Fig. S37).

The fabrication of the hydrogel electrode started with sequentially mixing the monomer (DMAPS, Macklin), crossed linker (Methylene-Bis-Acrylamide, MBA, Macklin), photo-initiator (α-Ketoglutaric acid, KGA, Macklin), ionic salt (LiCl, Macklin), and deionized water at a weight of 1833:2:1:400:3333. All materials were purchased from Aladdin. Next, the 2.83 mL AgNW solution (5 mg mL$^{-1}$ in isopropyl alcohol, Hengqiu Tech.) was added to the obtained hydrogel precursor, followed by a continuous string for 2 h. The PDMS mesh was prepared by engraving a 100 μm-thick PDMS film with the UV laser at a pulse frequency of 50 kHz and speed of 300 mm/s for 20 times repeated cutting. After dissolving the water-soluble tape to expose the copper electrode, the PDMS mesh was placed and the hydrogel precursor solution was poured, followed by curing in the UV pot at 30 W for 120 min, to form hydrogel-interfaced electrodes.

The fabrication of the LTCC antenna first used a punching machine (XT0800X) to punch holes (radius = 0.07 mm) in the ceramic germinal substrate at a speed of 1000 holes min$^{-1}$ (Fig. S15). Next, the conductive copper slurry (General Research Institute for Non-ferrous Metals) dispersed in these holes was sintered on the heater at 150 °C for 3 h. The top and bottom antennas were then printed on the germinal substrate through a laser-engraved mask. After peeling off the mask, sintering of the antenna at 150 °C for 1 h was followed by blading the top/bottom packaged ceramic and sintering at 870 °C for 1.5 h.

### Design of signal processing and transmission circuits
As illustrated in the detailed circuit schematic (Fig. S38), the signal processing unit included the low-power processor with 8-bit computing ability (Atmel, atmega328p), the inertial accelerometer (ANALOG, ADXL345), and the biological signal detection chip (NeuroSky, BMD101). The power management chip HT7133 (HOLTEK) was chosen due to its ability to convert the voltage from 3.7 to 3.3 V. The Bluetooth module PW02 (Phangwei Link) had serial port transmission ability. The integrated device can maintain continuous operation for approximately 5 to 6 h on a 35 mA h Li battery.

### FEA of mechanics and electromagnetics
The FEA was carried out to study the mechanical behaviors of the hydrogel/patch under diverse deformations and the EM properties of the LTCC antenna. All material properties were assigned according to the material source in the FEA software. The Young's modulus of these materials, the modified hydrogel, copper, eco-flex, polyimide, and skin replica, were set as 50 kPa, 80 GPa, 60 kPa, 800 MPa, 200 kPa, respectively.

### Characterizations of mechanical/electrical and structural properties
All tensile tests were conducted by a mechanical testing machine (ZQ-990B, Zhi Qu). Material conductivity was measured with an LCR digital bridge meter (IM2536, HIOKI). A pair of electrodes with a size of 5.5 mm × 32 mm (or diameter of 18.36 mm for the commercial gel electrodes) separated by a distance of 40 mm on the forearm were used for the contact impedance test. The return loss of the LTCC antenna was measured by a vector network analyzer (SVA 1032X, SIGLENT). The electrophysiological signals were acquired by a multichannel tester (BL-420, TECHMAN, Chengdu). The morphologies of the materials were observed by the scanning electron microscope (HiVac, Apreo). The cell viability was tested by the MTT Assay microplate reader. The light transmittance was obtained by the solar film transmission meter (LS101, LinShang). HR and RR were obtained as the

subject performed the instructed behaviors (e.g., holding the breath or sitting still).

## Machine learning algorithms for classification and prediction

Continuous measurements from fourteen healthy human subjects and two patients (one with myasthenia gravis and the other with laryngeal cancer) were used as the training data and the measurements from another two volunteers (a male and a female) were used as testing data. First, all measured data were marked at different feature types and normalized to unify data dimensions. Randomly dividing each 2D data of the laryngeal feature into 100 sequences with a fixed length of 1000 data points (covering the entire test data of each feature such as swallowing, drinking, speaking, and coughing) ensured data reliability. Meanwhile, every label was coded via the one hot coding[83]. Next, the pretreated data were extracted through 8-weight blocks with 62 processing layers and fed into the classifier. The batch size in the learning model was set to 64 for extracting the feature. The iterative learning rate (LR) was calculated as

$$LR = \left(1 + 0.5\cos\left(x + \pi/epochs\right)\right)*(1 - iLR), \qquad (1)$$

where $x$ is the times at the corresponding training process, the epochs are iterative times during the training, and iLR is the initial LR ($1e^{-4}$). The iterative attenuation rate (AR) was calculated from (with the initial value set as 0.9)

$$AR_t = AR_{t-1} - LR*\left(\frac{m_t}{1 - \beta_1^t}\right)\bigg/\left(\varepsilon + \sqrt{v_t/(1 - \beta_2^t)}\right), \qquad (2)$$

where $\beta_1$ and $\beta_2$ are the first and second-order attenuation coefficients, respectively, $m_t$ is the biased first-moment estimate, $v_t$ is the biased second raw moment estimate, and $\varepsilon$ is the relative weight constant. Equation (2) was based on the optimized machine learning algorithm (i.e., Adaptive moment estimation, Adam). The motion/movement artifacts (e.g., from drinking while coughing) were also accounted for in the testing data from the two new human subjects (see details in *SI Note1*).

## Development of the APP and cloud-served interface

The app was programmed using java on the Android Studio platform. The cloud-served interface was designed by the software development of the Internet of Things section in Alibaba Cloud. The pathological degree at the cloud server interface was sorted into eight levels (from the best state I to the worst state VIII) according to the rehabilitative conditions of three processes: swallowing (S), drinking water (D), and talking (T). During laryngeal rehabilitation, various behaviors such as swallowing, talking, and drinking water are assessed to be either normal or abnormal for each behavior, so the combined evaluation results of the three representative behaviors form eight evaluation states (Table S4). Although this 8-degree rating is not currently used in clinical evaluation, it could provide insights into patient's conditions to help guide individualized rehabilitation in the future.

## Statistics and reproducibility

No data were excluded from the analyses. No statistical method was used to predetermine the sample size. The results presented in Figs. 2b–e, h, i, 3a, b, f–h 4a–d, f, 6g–j, and Supporting Figs. 9, 11–13, 21, 22, 25–27 were obtained after three independent experiments with similar results.

## Reporting summary

Further information on research design is available in the Nature Portfolio Reporting Summary linked to this article.

## Data availability

The authors declare that all data supporting the results in this study are present in the paper and the data sources are available at https://doi.org/10.6084/m9.figshare.24311605. All other data supporting the findings of this study are available within the article and its supplementary files. Any additional requests for information can be directed to, and will be fulfilled by, the corresponding authors. Source data are provided with this paper.

## Code availability

The analysis codes used in this study are available from the corresponding authors upon request. The codes used for training laryngeal behaviors and circuits are openly available on GitHub at https://github.com/Hongcxu/Actions-training, ref. 84.

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

## Acknowledgements
The authors would like to acknowledge the financial support provided by the National Natural Science Foundation of China (Nos. 62274140 and 11922215), State Key Laboratory of Mechanics and Control of Mechanical Structures (Nanjing University of Aeronautics and Astronautics) (Grant No. MCMS-E-0422G03), the Fundamental Research Funds for the Central Universities (JB210407), the National Key Research and Development Program (Grant No. 2022YFB3204800), the Key Research and Development Program of Shanxi (Program No. 2021GY-277), the Shenzhen-Hong Kong-Macau Technology Research Program (Type C, 202011033000145), and the Innovation Fund of Xidian University.

## Author contributions
H.C.X. and L.B.G. conceived the idea and designed all experiments. H.C.X., W.H.Z., Y.Z., D.Q.Z., L.W., Y.B.Y., R.D.X., Z.M.H., N.J.Z., Y.X.Q., K.L., C.T. and Q.G., conducted the experiments. H.C.X., W.H.Z. and L.B.G. analyzed the data. H.C.X. wrote the manuscript. J.Z., Y.J.W., L.B.G., H.Y.C. Y.L.Z., G.C., H.Y.Z. and J.Y.Y. revised the original draft. L.B.G., W.D.W., H.Y.C. and Y.L. supervised the study. All the authors discussed the results and revised the final manuscript. All the authors have approved the final version of the paper.

## Competing interests
The authors declare no Competing Interests.

## Additional information

[1]School of Mechano-Electronic Engineering, Xidian University, Xian 710071, China. [2]Department of Medical Electronics, School of Biomedical Engineering, Air Force Medical University, Xi'an 710032, China. [3]Department of Otolaryngology-Head and Neck Surgery, The Second Affiliated Hospital of Air Force Medical University, Xi'an 710032, China. [4]Pen-Tung Sah Institute of Micro-Nano Science and Technology, Xiamen University, Xiamen 361102, China. [5]Applied Mechanics Laboratory, Department of Engineering Mechanics, Tsinghua University, Beijing 100084, China. [6]Engineering Research Center of Molecular and Neuro Imaging, Ministry of Education, School of Life Science and Technology, Xidian University, Xi'an, Shaanxi 710126, China. [7]Department of Mechanical and Energy Engineering, Southern University of Science and Technology, Shenzhen 518055, China. [8]Department of Engineering Science and Mechanics, The Pennsylvania State University, University Park, PA 16802, USA. [9]Department of Mechanical Engineering, The University of Hong Kong, Pokfulam, Hong Kong 999077, Hong Kong SAR. ✉e-mail: wangwd@mail.xidian.edu.cn; Huanyu.Cheng@psu.edu; ylu1@hku.hk; lbgao@xmu.edu.cn

