## [Peer review file · Nature Communications]

REVIEWER COMMENTS

Reviewer #1 (Remarks to the Author):

Xu and coauthors reported a stretchable device capable of wireless monitoring of fine movements and cloud-processing for analysis of the collected data, showing the way toward the next-generation wearable healthcare device platform. The device comprises separate inertial devices and force sensors to monitor both vibrations and muscle activities. Flexible surface electromyogram electrodes combined with low-impedance hydrogel enables measurement of high-quality local electrical signals. The collected signals are processed and transmitted for further analysis and diagnosis. Although each module of the proposed system, such as wearable sensing device, soft skin-device interface, and software-based processing, has been somehow studied in previous reports, the current work thoroughly integrated the necessary elements of a wireless skin-interfaced system. The reviewer believes that this report can be a meaningful contribution to the field of soft bioelectronics. Therefore, the reviewer recommends publication of this manuscript in Nature Communications after proper revision.

Comment #1. The reviewer thinks that mechanical durability of the laryngeal patch is not verified enough. Normally, the soldering region between the chips and Cu circuits is easily stressed and disconnected under external strain, so that it is required to examine the performance when stretched. Also, the authors need to evaluate if there is a performance drop after the cyclic stretching test.

Comment #2. The interface between the composite hydrogel and human skin was thoroughly researched. However, there is no analysis about the interface between a hydrogel and a metal electrode array. The reviewer recommends adding data and/or image of the hydrogel-metal interface and proper discussion.

Comment #3. Multiple motions occur simultaneously in real-world situations, and thus analyzing each motion selectively is essential for the practical use. The reviewer wonders if the proposed system could classify multiple motions separately with high precision (e.g., talking while walking, drinking water while coughing). If so, it needs to be demonstrated.

Comment #4. How is the VIA hole in Fig. S13. fabricated? It needs to be explained in the revised manuscript.

Comment #5. The fabrication process and figure images show that the open pad of the electrode directs to the opposite side to the skin (Fig. S14). Why is the device attached to the wrong side?

Comment #6. The concept of machine learning-assisted human activity analysis, including speech recognition, has been already suggested by several groups. In comparison to previous research, the authors collected sEMG and acceleration data, transferred them into 2D vectors, and classified into various motion/speech features. However, the advantage of integrating two distinct data is not highlighted enough. The reviewer recommends adding data or table that emphasizes the difference with previous studies, such as improved accuracy or detecting motions that were challenging to classify.

Reviewer #2 (Remarks to the Author):

What are the noteworthy results?

This manuscript has presented an interesting wearable sensor solution for remote monitoring of Post-surgical treatments monitoring applications. This kind of solution will be attractive to the digital healthcare domain. However, the main weakness of the manuscript is lack of a large range of testing

for various situation, which makes the results less convincing.

Will the work be of significance to the field and related fields? How does it compare to the established literature? If the work is not original, please provide relevant references.

The significance of the work was not very clear defined in the manuscript, and because this work contains a multi-disciplinary research, each part of innovation was not very clearly stated. For example, regarding the machine learning algorithm, only a well-known module was used, and no specific improvements were made for the proposed wearable sensors. Regarding the circuit design, they were also very basic, not sure what the main challenges to bring those on the skin wearable device, where the majority of electronic components were standard chip. One of the main concern is the power sources, as it can be seen from the schematics, the battery needs externally wired, which significantly reduces the compact design of the sensors. Flexible electronics have been research and report in the following paper, <https://pubs.acs.org/doi/10.1021/acs.accounts.8b00500>, authors focus on how this work in advances to the existing research regarding the improvements in each aspect of the design.

Does the work support the conclusions and claims, or is additional evidence needed?

Yes, some claims partially evidenced, however, due to limitation of testing sample sizes, it would be difficult to conclusion the scientific meanings for clinical applications.

Are there any flaws in the data analysis, interpretation and conclusions? Do these prohibit publication or require revision?

the data analysis was very limited, due to the size of samples, only 5 people were tested, and activities performed were very devised, therefore the classification tasks can be performed easily. One of the main challenges in the rehabilitation process, is to monitor how improvements of certain activities can be detected and classified, so track record or similar clinical research can be made accordingly. the work looks promise, but it is still at early stage.

Is the methodology sound? Does the work meet the expected standards in your field?

methodology looks ok, however, as stated early comments, authors should be focused on testing the platform in larger size of groups, including real patients, for the specific rehabilitation work, evidence the advances of using the proposed wearable sensor, and exam the accuracy.

Is there enough detail provided in the methods for the work to be reproduced?

no really, as the limitation of data sizes, detailed evaluation of using the proposed sensors, should be further elaborated.

Reviewer #3 (Remarks to the Author):

Thank you for the opportunity to review the paper titled "A fully integrated, standalone stretchable device platform with in-person adaptive machine learning for rehabilitation." This manuscript reports the design and validation process of a new wireless sensor. This reviewer applauds the authors on their work in this important area. However, this reviewer's enthusiasm for the work is lowered by a few concerns.

- The authors would benefit from working with a swallowing specialist as there are some areas that throughout the manuscript that requires input from a swallowing specialist. For example, on page 3, line 49, the authors use the words "genetic choking." It is not clear what genetic choking means. Furthermore, swallowing disorders can manifest as a result of many disorders including neurological disorders. It is not only related to "neck cancer or post-surgical treatment." Similarly, the title of the section on page 8, line 193 is post-surgical rehabilitation. However, post-surgical patients represent only part of the patients who receive swallowing rehabilitation. Furthermore, authors should also rephrase the sentence in lines 194-196 as it is not accurate. In addition, the difference between

drinking water and swallowing is not clear to this reviewer.

- Page 2, Line 33: The suprasternal notch refers to a specific anatomic location (the notch above the sternum). The location that the sensor was placed is not the suprasternal notch area.
- Page 3, lines 66-68: Authors refer to flexible EMGs that were developed (22, 24), however, mention that these suffer from low signal quality, motion artifacts, poor system integration, and severe skin inflammation. This reviewer recommends reviewing the citations 22 and 24 and reporting the findings accurately.
- The main concern about this paper is in the rationale (71-76). The authors report that there are skin-mounted mechano-acoustic sensors that track activity at the suprasternal notch and that high-quality sEMG signals have not been integrated into these platforms. While this is true, it is important to note that the suprasternal notch area or the laryngeal location that the authors mounted the device are not ideal locations to monitor swallowing activity. It has been well established that larynx moves throughout the swallow including before and after the swallow because of motion unrelated to swallowing. For example, chewing results in the inferior and superior movements of the larynx that are similar to the movements observed during swallowing. Furthermore, it has been well established through intramuscular electrodes, surface electrodes and imaging technologies that the ideal location for capturing swallowing movements using sEMG is the suprahyoid area. This reviewer is concerned about the location of the sensor and the integrity of the signals that were obtained using the sensors. I recommend the authors to use imaging technologies such as VFSS (the current gold-standard method) to validate their signals and ensure that the signals that they are reporting are actually swallowing signals.
- The authors report data from 2 healthy adults. However, they also make claims such as “to help evaluate the post-surgical state...” page 10, line 233. It is not clear to this reviewer how authors are able to make these generalizations by using the data obtained from healthy individuals who did not receive head and neck surgery.
- Page 15, lines 356-361: The authors report that the pathological degree was sorted into eight levels. Are the authors referring to severity? If so, it would be important to clarify how these 8 levels were determined for swallowing, drinking water, and talking. This reviewer is not familiar with any 8-level evidence-based severity ratings related to swallowing and speech impairments.

Reviewer #1 (Remarks to the Author):

Xu and coauthors reported a stretchable device capable of wireless monitoring of fine movements and cloud-processing for analysis of the collected data, showing the way toward the next-generation wearable healthcare device platform. The device comprises separate inertial devices and force sensors to monitor both vibrations and muscle activities. Flexible surface electromyogram electrodes combined with low-impedance hydrogel enables measurement of high-quality local electrical signals. The collected signals are processed and transmitted for further analysis and diagnosis. Although each module of the proposed system, such as wearable sensing device, soft skin-device interface, and software-based processing, has been somehow studied in previous reports, the current work thoroughly integrated the necessary elements of a wireless skin-interfaced system. The reviewer believes that this report can be a meaningful contribution to the field of soft bioelectronics. Therefore, the reviewer recommends publication of this manuscript in Nature Communications after proper revision.

Our response: We highly appreciate the reviewer's positive evaluation of our work. We also appreciate the insightful comments that help significantly improve the overall quality of this work.

Comment #1. *The reviewer thinks that mechanical durability of the laryngeal patch is not verified enough. Normally, the soldering region between the chips and Cu circuits is easily stressed and disconnected under external strain, so that it is required to examine the performance when stretched. Also, the authors need to evaluate if there is a performance drop after the cyclic stretching test.*

Our response: Many thanks for the comments. We have investigated the resistance variation of one representative soldering region between the chips and Cu circuits (along the stretching direction) under 30% stretching (**Fig. S6a**). The measured resistance is the sum of the chip resistance ($R_0=1\text{ k}\Omega$) and the resistance from the soldering region (ΔR) (**Fig. S6b**). The resistance variation of 0.8% (with extremely low power consumption) is negligibly small after

1500 cycles for 30% uniaxial stretching (Fig. S6c).

Fig. S6. Stretching test at the representative soldering region. **a** Optical image of the integrated device with the representative soldering region connected to a packaged resistor ($1\text{ k}\Omega$) shown in the inset. **b** Equivalent circuit of the soldering region. **c** Long-term durability test over 1500 cycles for stretching of 30%.

The integrated device also operates without any issues with the indicator LED on during uniaxial stretching of 30% and cyclic stretching of 25% over 3600 cycles due to the small strain in the device as verified by the FEA (Fig. R1 and movie S3).

Fig. R1. Stretching tests of the integrated patch. **a** Optical image of the integrated patch for 30% uniaxial stretching. **b** Strain distribution in the encapsulation silicone film and **c** serpentine network under 25% stretching from FEA. **d** Long-term durability test of the integrated device over 3600 cycles under a stretching of 25% with the optical image of the patch after the test shown in the inset.

Our revision to the manuscript:

The standalone stretchable device platform fabricated from low-cost processes (**Figs. S2-5**) exhibits robust **electromechanical** performance upon various mechanical deformations (e.g., stretching, bending, and twisting) as verified by both finite element analysis (FEA) and experiments (**Figs. 1e, S6, and movies S1-3**).

Comment #2. The interface between the composite hydrogel and human skin was thoroughly researched. However, there is no analysis about the interface between a hydrogel and a metal electrode array. The reviewer recommends adding data and/or image of the hydrogel-metal interface and proper discussion.

Our response: We thank the reviewer for this suggestion. We have measured the resistance at the interface between the hydrogel and Cu-electrode array (**Fig. S12**), which is comparable to the pristine hydrogel.

Our revision to the manuscript: We have included the new experimental results in the revised manuscript.

“The lower contact impedance results from **the high conductivity at the Cu/hydrogel interface (Fig. S12)** and the improved hydrogel/skin contact quality as observed on the skin replica (**Fig. 2f**).”

Fig. S12. Resistance at the interface between the Cu-mesh electrode and the composite hydrogel over the frequency range from 4 Hz to 100 kHz, demonstrating high conductivity at the interface.

Comment #3. Multiple motions occur simultaneously in real-world situations, and thus analyzing each motion selectively is essential for the practical use. The reviewer wonders if the proposed system could classify multiple motions separately with high precision (e.g., talking while walking, drinking water while coughing). If so, it needs to be demonstrated.

Our response: Many thanks for the suggestion. We have carried out additional experiments to show that our system can distinguish multiple motions separately (e.g., talking while walking, drinking water while coughing) owing to the simultaneously measured acceleration data along different directions. The z-axial acceleration is mostly associated with talking and negligible during walking (Fig. S21a), whereas the x-axial acceleration is dominated by walking.

Additionally, the x-axis acceleration can be used to identify drinking water, whereas coughing shows large magnitude in the z- and y-axis accelerations (**Fig. S21b**).

Fig. S21. Triaxial accelerations captured by the stretchable patch at the throat to identify a talking while walking and b drinking water while coughing.

Our revision to the manuscript: We have included the new measurements and corresponding discussion in the revised manuscript.

“The high-sensitivity accelerometer ADXL-345 with a sampling frequency of up to 800 Hz in the patch allows successful continuous monitoring of various activities, such as sitting, talking, swallowing, walking, and jumping (**Fig. 4a** and **movie S4-6**), with a wide frequency spectrum from 0 to 400 Hz (**Fig. 4b**). The simultaneously measured acceleration data along three different directions allow the integrated system to distinguish multiple motions separately (e.g., talking while walking, drinking water while coughing, drinking water while swallowing) (**Figs. S21** and **S22**), as well as capture the swallowing process from a patient with myasthenia gravis (**Fig. S23**).”

Comment #4. How is the VIA hole in **Fig. S13**. fabricated? It needs to be explained in the revised manuscript.

Our response: We thank the reviewer for this suggestion. We have included the fabrication process of the VIA hole in the revised manuscript (**Fig. S15**, previous **Fig. S13**).

Our revision to the manuscript: We have included the fabrication process of the VIA hole in the Methods section of the revised manuscript (**Fig. S15**).

“After dissolving the water-soluble tape to expose the copper electrode, the PDMS mesh was placed and the hydrogel precursor solution was poured, followed by curing in the UV pot at 30 W for 120 min, to form hydrogel-interfaced electrodes.

The fabrication of the LTCC antenna first used a punching machine (XT0800X) to punch holes (radius = 0.07 mm) in the ceramic germinal substrate at a speed of 1000 holes min⁻¹ (**Fig. S15b**). Next, the conductive copper slurry dispersed in these holes was sintered on the heater at 150°C for 3 h. The top and bottom antennas were then printed on the germinal substrate through a laser-engraved mask. After peeling off the mask, sintering of the antenna at 150°C for 1 h was followed by blading the top/bottom packaged ceramic and sintering at 870°C for 1.5 h.”

Fig. S15. Fabrication of the low-temperature co-fired ceramic (LTCC) antenna. a Design and b fabrication process of the antenna.

Comment #5. *The fabrication process and figure images show that the open pad of the electrode directs to the opposite side to the skin (Fig. S14). Why is the device attached to the wrong side?*

Our response: We thank the reviewer for the careful check. Indeed, the device was attached to the skin with an opposite side so the device components are facing outside for direct and easy viewing (Fig. S16, previous Fig. S14). We have updated the figure caption to avoid confusion.

Our revision to the manuscript: We have updated the figure caption to avoid confusion.

Fig. S16. Optical and thermal images of the patch on the laryngeal skin. **a** Optical image of the patch on the skin. Thermal images to show the comparison in the temperature distribution of the powered patch on **b** a cooled platform (10°C) and on the skin in the **c** front and **d** side views.

***Comment #6.** The concept of machine learning-assisted human activity analysis, including speech recognition, has been already suggested by several groups. In comparison to previous research, the authors collected sEMG and acceleration data, transferred them into 2D vectors, and classified into various motion/speech features. However, the advantage of integrating two distinct data is not highlighted enough. The reviewer recommends adding data or table that emphasizes the difference with previous studies, such as improved accuracy or detecting motions that were challenging to classify.*

Our response: We appreciate for the reviewer’s comments. Comparison in the overall accuracy between our work and previous reports indicates the advantage of integrating two distinct signal inputs in improved prediction accuracy (**Table S3**).

Table S3 Comparison of the advantage of the integration of two signals with reported methods

Applications	Signal type	Data dimension	Number of Signal	Overall accuracy	Reference
Facial strain and kinematics	Strain	1	1	86.8%	30
Throat activities	Resistance	1	1	92.73%	71
Speech signal	sEMG	1	1	92.3%	73
Speech detection	sEMG	4	1	87.53%	76
Gesture recognition	sEMG	1	1	92.87%	78
Non-invasive identification	sEMG	1	1	92.5%	79
Finger motion	sEMG	4	1	<80%	80
Gesture recognition	Piezoelectric and triboelectric	2	2	82.3%	81
Object recognition	Pressure and temperature	10	2	94%	82
Laryngeal rehabilitation	sEMG and acceleration	4	2	98.2%	This work

sEMG Surface Electromyogram

Our revision to the manuscript: We have compared the overall accuracy of our work with the previous literature reports to highlight the advantage of integrating two distinct signal inputs.

“To help automatically evaluate the laryngeal condition from the new patients and healthy individuals, a CNN-based 2D-like sequential feature extractor (2D-SFE) is explored to classify and infer pathological status based on the classification of physiological events (**Fig. 5a**).

Comparison in the overall accuracy between this work and previous reports indicates the advantage of integrating two distinct signal inputs in adaptive machine learning^{30, 68, 70, 73, 75-79}(**Table S3**).”

References:

71. Gong, S. et al. Hierarchically resistive skins as specific and multimetric on-throat wearable biosensors. *Nat Nanotechnol* (2023).
73. Liu, H. et al. An epidermal sEMG tattoo-like patch as a new human-machine interface for patients with loss of voice. *Microsyst Nanoeng* **6**, 16 (2020).
76. Tao, L.Q. et al. An intelligent artificial throat with sound-sensing ability based on laser induced graphene. *Nat Commun* **8**, 14579 (2017).
78. Moin, A. et al. A wearable biosensing system with in-sensor adaptive machine learning for hand gesture recognition. *Nature Electronics* **4**, 54–63 (2021).
79. Clarke, A.K. et al. Deep Learning for Robust Decomposition of High-Density Surface EMG Signals. *IEEE Trans Biomed Eng* **68**, 526-534 (2021).
80. Lee, H., Kim, D. & Park, Y.L. Explainable Deep Learning Model for EMG-Based Finger Angle Estimation Using Attention. *IEEE Trans Neural Syst Rehabil Eng* **30**, 1877-1886 (2022).
81. Syu, M.H., Guan, Y.J., Lo, W.C. & Fuh, Y.K. Biomimetic and porous nanofiber-based hybrid sensor for multifunctional pressure sensing and human gesture identification via deep learning method. *Nano Energy* **76**, 105029 (2020).
82. Li, G., Liu, S., Wang, L. & Zhu, R. Skin-inspired quadruple tactile sensors integrated on a robot hand enable object recognition. *Sci Robot* **5**, eabc8134 (2020).

Reviewer #2 (Remarks to the Author):

What are the noteworthy results? This manuscript has presented an interesting wearable sensor solution for remote monitoring of Post-surgical treatments monitoring applications. This kind of solution will be attractive to the digital healthcare domain. However, the main weakness of the manuscript is lack of a large range of testing for various situation, which makes the results less convincing.

Our response: We appreciate the referee's positive evaluation of this work. Many thanks for the reviewer's comments and suggestions that help significantly improve the overall quality of this work. We agree that the manuscript would benefit from a large range of testing, so we have included new results from another 11 subjects (including two patients with the myasthenia gravis and laryngeal cancer) for throat rehabilitation, as in the detailed reply for **Comments #3-6**.

Comment #1. *Will the work be of significance to the field and related fields? How does it compare to the established literature? If the work is not original, please provide relevant references.*

Our response: Many thanks for the comment. The developed flexible sensing electronic platform integrates the soft sensors for biophysical monitoring with a stretchable hybrid circuit for wireless transmission, which provides new insights for evaluation and rehabilitation of swallowing disorders. Compared with the previous reports [Sensors and Actuators A: Physical 295(2019):678-686, IEEE J Biomed Health Inform (2017):1-1], the hydrogel-based sensors feature significantly reduced contact impedance for motion-artifact free sensing of sEMG. In addition, most of the current laryngeal devices only use one resistive sensor to measure on-throat signatures such as voice, swallowing, and neck movement (**Fig. R2**) [Gong, S. et al. Nat. Nanotechnol (2023), Nat. Mach. Intell 5(2): 169-80(2023), and Nat Commun 8, 14579 (2017)], which are associated with limited information and sometimes poor signal quality. In comparison, the sEMG combined with 3D acceleration signals captured by our integrated device platform can provide rich information for highly precise recognition and evaluation, outperforming previous

reported laryngeal sensors (**Table S2**). Comparison in the overall accuracy between our work and previous reports indicates the advantage of integrating two distinct signal inputs in improved prediction accuracy (**Table S3**).

Fig. R2. Previously reported individual on-throat sensors or electrodes without the integrated circuit.

We have carried out additional experiments to show that our system can distinguish multiple motions separately (e.g., talking while walking, drinking water while coughing) owing to the simultaneously measured acceleration data along different directions. The z-axial acceleration is mostly associated with talking and negligible during walking (**Fig. S21a**), whereas the x-axial acceleration is dominated by walking. Additionally, the x-axis acceleration can be used to identify drinking water, whereas coughing shows large magnitude in the z- and y-axis accelerations (**Fig. S21b**).

Fig. S21. Triaxial accelerations captured by the stretchable patch at the throat to identify a talking while walking and b drinking water while coughing.

Table S2 Comparison of different laryngeal sensor patches

Sensor patch	Flexibility	Systemic integration	Wireless	Sensing mode	Monitoring	Trained by ML	Reference
Flexible submental sensor	Semi-flexible	N	√	Strain, EMG	Muscle activity, throat movements	N	23
Accelerometer Sensor	Semi-flexible	N	N	Acceleration	Voices	N	24
Mechano-acoustic device	Semi-flexible	√	√	Acceleration	Physiological processes, body motions	N	38
Acoustic MEMS Sensor	/	N	N	Acoustic vibration	Chewing, Swallowing	√	63
Epidermal device	Full-soft	N	N	Acceleration, EMG	Electrophysiological signals	√	64
Dual-axis swallowing accelerometer	Semi-flexible	N	N	Acceleration	Swallowing	N	65
Dual wearable sensor	Semi-flexible	√	√	Acceleration	Artifact-canceled physiological	N	66
Mechano-acoustic sensors	Semi-flexible	√	√	Acceleration	Swallowing, respirations	N	67
Throat vibrator	Rigid	N	N	Microphone vibration	Dysphagia	√	68
Lip-closing force gauge	Rigid	N	N	Force, EMG	Lip-closing	N	16
Neck belt	Semi-flexible	N	N	Displacements	Swallowing	N	69
Acoustic detecting array	Semi-flexible	N	N	Acceleration, microphone	Swallowing sounds	N	70
Hierarchically resistive skins	Full-soft	N	N	Resistive changes	Touch and neck movement	√	71
Wearable artificial throat	Full-soft	N	N	Strain	Voices	√	72
sEMG tattoo	Full-soft	N	N	sEMG	Speech	N	73
2D Metal Film	Full-soft	N	N	Resistive changes	Subvocal talking	N	74
Intelligent artificial throat	Full-soft	N	N	Strain, EMG	Voice signal	√	75
Laser-induced	Semi-	N	N	Resistive	Sound signal	N	76

graphene sensor	flexible			changes			
Proposed sensor patch	Full-soft	√	√	sEMG, Acceleration	Swallowing, dysphagia	√	This work
N no, V yes, EMG Electromyogram, sEMG Surface Electromyogram, ML: Machine-learning							

Table S3 Comparison of the advantage of the integration of two signals with reported methods

Applications	Signal type	Data dimension	Number of Signal	Overall accuracy	Reference
Facial strain and kinematics	Strain	1	1	86.8%	30
Throat activities	Resistance	1	1	92.73%	71
Speech signal	sEMG	1	1	92.3%	73
Speech detection	sEMG	4	1	87.53%	76
Gesture recognition	sEMG	1	1	92.87%	78
Non-invasive identification	sEMG	1	1	92.5%	79
Finger motion	sEMG	4	1	<80%	80
Gesture recognition	Piezoelectric and triboelectric	2	2	82.3%	81
Object recognition	Pressure and temperature	10	2	94%	82
Laryngeal rehabilitation	sEMG and acceleration	4	2	98.2%	This work
sEMG Surface Electromyogram					

Our revision to the manuscript: We have included additional discussion:

“The high-sensitivity accelerometer ADXL-345 with a sampling frequency of up to 800 Hz in the patch allows successful continuous monitoring of various activities, such as sitting, talking, swallowing, walking, and jumping (**Fig. 4a** and **movies S4-6**), with a wide frequency spectrum from 0 to 400 Hz (**Fig. 4b**). **The simultaneously measured acceleration data along three different directions allow the integrated system to distinguish multiple motions separately (e.g., talking while walking, drinking water while coughing, drinking water while swallowing) (Figs. S21 and S22), as well as capture the swallowing process from a patient with myasthenia gravis (Fig. S23).**”

“To help automatically evaluate the laryngeal condition **from the new patients and healthy individuals**, a CNN-based 2D-like sequential feature extractor (2D-SFE) is explored to classify and infer pathological status based on the classification of physiological events (**Fig. 5a**).

The fully integrated device with superior mechanical and electrical properties for conformable monitoring (e.g., swallowing activities) during movements outperforms previous reports of laryngeal sensors ^{16, 23, 24, 34, 60-73} (**Table S2**)”

Comparison in the overall accuracy between this work and previous reports indicates the advantage of integrating two distinct signal inputs in adaptive machine learning ^{30, 68, 70, 73, 75-79} (Table S3).

References:

71. Gong, S. et al. Hierarchically resistive skins as specific and multimetric on-throat wearable biosensors. *Nat Nanotechnol* (2023).
72. Yang, Q. et al. Mixed-modality speech recognition and interaction using a wearable artificial throat. *Nature Machine Intelligence* **5**, 169-180 (2023).

73. Liu, H. et al. An epidermal sEMG tattoo-like patch as a new human-machine interface for patients with loss of voice. *Microsyst Nanoeng* **6**, 16 (2020).
74. Qin, R. et al. Protein-Bound Freestanding 2D Metal Film for Stealth Information Transmission. *Adv Mater* **31**, e1803377 (2019).
75. Qiao, Y. et al. Electromyogram-strain synergetic intelligent artificial throat. *Chem Eng J* **449**, 137741 (2022).
76. Tao, L.Q. et al. An intelligent artificial throat with sound-sensing ability based on laser induced graphene. *Nat Commun* **8**, 14579 (2017).
77. Crary, M.A. & Carnaby, G.D. Adoption into clinical practice of two therapies to manage swallowing disorders: exercise-based swallowing rehabilitation and electrical stimulation. *Curr Opin Otolaryngo* **22**, 172-180 (2014).
78. Moin, A. et al. A wearable biosensing system with in-sensor adaptive machine learning for hand gesture recognition. *Nature Electronics* **4**, 54–63 (2021).
79. Clarke, A.K. et al. Deep Learning for Robust Decomposition of High-Density Surface EMG Signals. *IEEE Trans Biomed Eng* **68**, 526-534 (2021).
80. Lee, H., Kim, D. & Park, Y.L. Explainable Deep Learning Model for EMG-Based Finger Angle Estimation Using Attention. *IEEE Trans Neural Syst Rehabil Eng* **30**, 1877-1886 (2022).
81. Syu, M.H., Guan, Y.J., Lo, W.C. & Fuh, Y.K. Biomimetic and porous nanofiber-based hybrid sensor for multifunctional pressure sensing and human gesture identification via deep learning method. *Nano Energy* **76**, 105029 (2020).
82. Li, G., Liu, S., Wang, L. & Zhu, R. Skin-inspired quadruple tactile sensors integrated on a robot hand enable object recognition. *Sci Robot* **5**, eabc8134 (2020).

Comment#2. *The significance of the work was not very clear defined in the manuscript, and because this work contains a multi-disciplinary research, each part of innovation was not very*

clearly stated. For example, regarding the machine learning algorithm, only a well-known module was used, and no specific improvements were made for the proposed wearable sensors. Regarding the circuit design, they were also very basic, not sure what the main challenges to bring those on the skin wearable device, where the majority of electronic components were standard chip. One of the main concern is the power sources, as it can be seen from the schematics, the battery needs externally wired, which significantly reduces the compact design of the sensors. Flexible electronics have been research and report in the following paper, <https://pubs.acs.org/doi/10.1021/acs.accounts.8b00500>, authors focus on how this work in advances to the existing research regarding the improvements in each aspect of the design.

Our response: We thank the reviewer for this comment. Because of superior generalization of the CNN, the proposed feature extractor in the machine learning algorithm can adapt to new subjects rather than merely classifying various signatures as in previous reports [Gun-Hee Lee, et al, ACS nano 14.9(2020). Kim, K.K., Ha, I., Kim, M. et al. Nat Commun 11, 2149 (2020)]. Furthermore, the sEMG combined with 3D acceleration signals captured by our integrated device platform can provide rich information for highly precise recognition and evaluation, outperforming previous reported laryngeal sensors (**Table S2**).

Although the battery as the power source needs to be connected through the serpentine wires, the Li battery with a capacity of 35 mA h can be well integrated in the stretchable circuit and encapsulated with a white Ecoflex (**Fig. R3**) to continuously drive the system for 5-6 hours. Wireless charging of the battery is also possible to further extend the operation time [Xu, S., Zhang, Y., Cho, J. et al. Nat Commun 4, 1543 (2013)].

Fig. R3. Optical images of the integrated device system with the battery **a** before and **b** after encapsulation with white Ecoflex and **c** its conformal contact with the hand back.

Our revision to the manuscript: We have discussed the novelty and advantages of our integrated device system in the context of literature reports in the revised manuscript:

“The fully connected neuron in the machine learning model further allows it to accurately classify the data from new human subjects with a 92% accuracy. In addition, the custom-built interface to process the data on the cloud opens up new opportunities for remote diagnosis and treatment evaluation for rehabilitation management and various disease applications. Above all, **the developed standalone stretchable platform integrates the skin-interfaced soft electrodes and sensors for biophysical monitoring with a stretchable hybrid circuit for wireless transmission. Furthermore, the sEMG combined with 3D acceleration signals captured by our integrated device platform can provide rich information for the feature extractor to adapt to new human subjects, allowing evaluation and rehabilitation of swallowing disorders.**”

“Wearable devices start to gain momentum in **disease diagnostic confirmation, treatment evaluation, and healthy aging**¹⁻⁴.”

Reference:

4. Gao, W., Ota, H., Kiriya, D., Takei, K. & Javey, A. Flexible Electronics toward Wearable Sensing. *Acc Chem Res* 52, 523-533 (2019).

Comment #3. Does the work support the conclusions and claims, or is additional evidence needed? Yes, some claims partially evidenced, however, due to limitation of testing sample sizes, it would be difficult to conclusion the scientific meanings for clinical applications.

Our response: Many thanks for the comment. We have carried out testing of the device with another 11 human subjects (including two patients who suffer from the myasthenia gravis and the laryngeal cancer, respectively).

The swallowing process with the pear-shaped postcricoid area transitioned from opening to closing can be accurately monitored by our integrated wireless platform, which is also validated against the gold standard based on the fiberoptic endoscopic examination of swallowing (FEES). Compared with the healthy control (**Fig. S34a**), the patient with myasthenia gravis shows higher muscle force to complete swallowing (**Fig. S34b**, green dashed ellipse). In comparison, the FEES is not able to capture this difference.

Fig. S34. Comparison between a healthy control and b patient with myasthenia gravis during swallowing tests captured by our integrated wireless platform and the FEES: Real-time screenshot on the App interface (left), image showing the device at the oropharyngeal junction area and the FEES into the esophagus (middle), and FEES images of the closing process of the pear-shaped postcricoid area (right).

The volume and viscosity swallowing test-chinses version (VVST-CV) with modified safety and

effectiveness indicators is used to quantify the analysis as the subject ingests a blue edible indicator with a body weight percentage of 2% (stage #1) and 1% (stage #2) (**Fig. S35a**). The clinical standard for swallowing rhythm requires swallowing to be less than 3 s from the the indicator in mouth to the circumpharyngeal muscle opening and a total time of less than 15 s for the complete ingestion process. The swallowing in both stages for the patient with myasthenia gravis exceeds 15 s (**Fig. S35b**). Liquid residual still exists after 25 s in the oropharyngeal junction area in stage #1 (**Fig. S35c**). The duration of the process of about 10 s captured by our device is consistent with the clinical standard (**Fig. S35d**), demonstrating the feasibility and reliability of the integrated system. It is important to note that one portion of the ingesting process (7-12s) in the stage #2 is not captured by the FEES due to the closed oropharyngeal junction area to block the endoscopic view (**Fig. S35e**). In contrast, this missing swallowing process (green dashed ellipse) is still clearly recorded by our device (**Fig. S35f**), providing enhanced diagnostic accuracy for laryngeal postoperative patients.

Fig. S35. Volume and viscosity swallowing test with modified safety and effectiveness indicators (VVST-CV) for the patient with myasthenia gravis. **a** Optical images showing the patient swallowing a blue edible indicator with a weight rate of 2% (left, stage #1) and 1% (right, stage #2). **b** Real-time signals captured by our device during the two-stage swallowing. **c** FEES images in stage #1 and **d** the corresponding time-interval as a function of the number of swallowing. **e** FEES images in stage #2 and **f** the time-interval as a function of number of swallowing.

Furthermore, the device is also applied to record the swallowing process for an old patient with a recent operation for the laryngeal cancer (Fig. R4). The leaked liquid (to the trachea) and residual at the oropharyngeal junction area result in an abnormal swallowing process (incomplete swallowing of the blue indicator due to with severe coughing) with longer duration for each

swallowing/ingestion and severe coughing.

Fig. R4. VVST-CV for an old patient with a clinical operation for laryngeal cancer. Optical image of the patient swallowing a blue edible indicator (left), real-time data displayed on the APP (middle), and FEES images during swallowing (right).

We have also carried out additional experiments to show that our system can distinguish multiple motions separately (e.g., talking while walking, drinking water while coughing) owing to the simultaneously measured acceleration data along different directions. The z-axial acceleration is mostly associated with talking and negligible during walking (**Fig. S21a**), whereas the x-axial acceleration is dominated by walking. Additionally, the x-axis acceleration can be used to identify drinking water, whereas coughing shows large magnitude in the z- and y-axis accelerations (**Fig. S21b**).

Fig. S21b. Triaxial accelerations captured by the stretchable patch at the throat to identify a talking while walking and b drinking water while coughing.

Our modification to the manuscript: We have included the new experimental results with relevant discussion in the revised manuscript.

“ The difference in the fundamental frequency (e.g., a higher value of over 100 Hz from talking and coughing compared with that of other events) allows the evaluation of various laryngeal conditions.

The swallowing process with the pear-shaped postcricoid area transitioned from opening to closing can be accurately monitored by our integrated wireless platform, which is also validated against the gold standard based on the fiberoptic endoscopic examination of swallowing (FEES, **Figs. S34-35** and **Movie S7**). Compared with the healthy control (**Fig. S34a**), the patient with myasthenia gravis shows higher muscle force to complete swallowing, where this difference is not captured by the FEES (**Fig. S34b**).

The volume and viscosity swallowing test-chinses version (VVST-CV) with modified safety and effectiveness indicators is used to quantify the analysis as the subject ingests a blue edible indicator with a body weight percentage of 2% (stage #1) and 1% (stage #2) (**Fig. S35a**). The clinical standard for swallowing rhythm requires swallowing to be less than 3 s from the indicator in mouth to the circumpharyngeal muscle opening and a total time of less than 15 s for the complete ingestion process. The swallowing in both stages for the patient with myasthenia gravis exceeds 15 s (**Fig. S35b**). Liquid residual still exists after 25 s in the oropharyngeal junction area in stage #1 (**Fig. S35c**). The duration of the process of about 10 s captured by our device is consistent with the clinical standard (**Fig. S35d**), demonstrating the feasibility and reliability of the integrated system. It is important to note that one portion of the ingesting process (7-12s) in the stage #2 is not captured by the FEES due to the closed oropharyngeal junction area to block the endoscopic

view (**Fig. S35e**). In contrast, this missing swallowing process is still clearly recorded by our device (**Fig. S35f**, green dashed ellipse), providing enhanced diagnostic accuracy for laryngeal postoperative patients.”

***Comment #4.** Are there any flaws in the data analysis, interpretation and conclusions? Do these prohibit publication or require revision? the data analysis was very limited, due to the size of samples, only 5 people were tested, and activities performed were very devised, therefore the classification tasks can be performed easily. One of the main challenges in the rehabilitation process, is to monitor how improvements of certain activities can be detected and classified, so track record or similar clinical research can be made accordingly. the work looks promise, but it is still at early stage.*

Our response: We thank the reviewer for this comment. We have carried out testing of the device with another 11 human subjects (including two patients who suffer from the myasthenia gravis and the laryngeal cancer, respectively).

The swallowing process with the pear-shaped postcricoid area transitioned from opening to closing can be accurately monitored by our integrated wireless platform, which is also validated against the gold standard based on the fiberoptic endoscopic examination of swallowing (FEES). Compared with the healthy control (**Fig. S34a**), the patient with myasthenia gravis shows higher muscle force to complete swallowing (**Fig. S34b**, green dashed ellipse). In comparison, the FEES is not able to capture this difference.

Fig. S34. Comparison between a healthy control and b patient with myasthenia gravis during swallowing tests captured by our integrated wireless platform and the FEES: Real-time screenshot on the App interface (left), image showing the device at the oropharyngeal junction area and the FEES into the esophagus (middle), and FEES images of the closing process of the pear-shaped postcricoid area (right).

The volume and viscosity swallowing test-chinses version (VVST-CV) with modified safety and effectiveness indicators is used to quantify the analysis as the subject ingests a blue edible indicator with a body weight percentage of 2% (stage #1) and 1% (stage #2) (**Fig. S35a**). The clinical standard for swallowing rhythm requires swallowing to be less than 3 s from the the indicator in mouth to the circumpharyngeal muscle opening and a total time of less than 15 s for the complete ingestion process. The swallowing in both stages for the patient with myasthenia gravis exceeds 15 s (**Fig. S35b**). Liquid residual still exists after 25 s in the oropharyngeal junction area in stage #1 (**Fig. S35c**). The duration of the process of about 10 s captured by our device is consistent with the clinical standard (**Fig. S35d**), demonstrating the feasibility and reliability of the integrated system. It is important to note that one portion of the ingesting process (7-12s) in the stage #2 is not captured by the FEES due to the closed oropharyngeal junction area to block the endoscopic view (**Fig. S35e**). In contrast, this missing swallowing process (green dashed ellipse) is still clearly recorded by our device (**Fig. S35f**), providing enhanced diagnostic accuracy for laryngeal

postoperative patients.

Fig. S35. Volume and viscosity swallowing test with modified safety and effectiveness indicators (VVST-CV) for the patient with myasthenia gravis. a Optical images showing the patient swallowing a blue edible indicator with a weight rate of 2% (left, stage #1) and 1% (right, stage #2). **b** Real-time signals captured by our device during the two-stage swallowing. **c** FEES images in stage #1 and **d** the corresponding time-interval as a function of the number of swallowing. **e** FEES images in stage #2 and **f** the time-interval as a function of number of swallowing.

Furthermore, the device is also applied to record the swallowing process for an old patient with a recent operation for the laryngeal cancer (Fig. R4). The leaked liquid (to the trachea) and residual at the oropharyngeal junction area result in an abnormal swallowing process (incomplete

swallowing of the blue indicator due to with severe coughing) with longer duration for each swallowing/ingestion and severe coughing.

Fig. R4. VVST-CV for an old patient with a clinical operation for laryngeal cancer. Optical image of the patient swallowing a blue edible indicator (left), real-time data displayed on the APP (middle), and FEES images during swallowing (right).

In addition, the swallowing signals over several repetitions from a patient with myasthenia gravis and 9 healthy subjects are captured and compared (Fig. S23), indicating higher muscle force (acceleration along z-axis) to complete swallowing in the patient with myasthenia gravis. Although the captured data can help build the individualized database of the laryngeal postoperative signature, the rehabilitation is out of scope of this work and it will be pursued in our future studies.

Fig. S23. Normalized swallowing signals of 9 healthy subjects and the patient with myasthenia gravis when they swallows with an repetition.

Our revision to the manuscript: We have included the new results in the revised manuscript.

“The high-sensitivity accelerometer ADXL-345 with a sampling frequency of up to 800 Hz in the patch allows successful continuous monitoring of various activities, such as sitting, talking, swallowing, walking, and jumping (**Fig. 4a** and **movie S4-6**), with a wide frequency spectrum from 0 to 400 Hz (**Fig. 4b**). The simultaneously measured acceleration data along three different directions allow the integrated system to distinguish multiple motions separately (e.g., talking while walking, drinking water while coughing, drinking water while swallowing) (**Figs. S21** and **S22**), as well as capture the swallowing process from a patient with myasthenia gravis (**Fig. S23**).”

Comment #5. Is the methodology sound? Does the work meet the expected standards in your field? methodology looks ok, however, as stated early comments, authors should be focused on testing the platform in larger size of groups, including real patients, for the specific rehabilitation work, evidence the advances of using the proposed wearable sensor, and exam the accuracy.

Our response: We appreciate for the reviewer’s comment and suggestion. We have carried out testing of the device with another 11 human subjects (including two patients who suffer from the myasthenia gravis and the laryngeal cancer, respectively). The swallowing signals over several repetitions from a patient with myasthenia gravis and 9 health adults are captured and compared (**Fig. S23**), indicating higher muscle force (acceleration along z-axis) to complete swallowing in the patient with myasthenia gravis. Although the captured data can help build the individualized database of the laryngeal postoperative signature, the rehabilitation is out of scope of this work and it will be pursued in our future studies.

Fig. S23. Normalized swallowing signals of 9 healthy subjects and the patient with myasthenia gravis when they swallow with a repetition.

Furthermore, the difference between drinking water and swallowing lies in the duration and signal patterns (Fig. S22). The process of drinking water consists of holding water in the mouth lasting for 840 ms, flowing along the throat for 150 ms with a large acceleration magnitude, and swallowing through the pear-shaped postcricoid area for 270 ms (Fig. S22a). In contrast, swallowing only exhibits swallowing through the pear-shaped postcricoid area for 490 ms with a relatively smaller acceleration (Fig. S22b).

Fig. S22. Comparison of the acceleration responses between a drinking water and b swallowing.

Our revision to the manuscript: We have included the new results in the revised manuscript.

“With a 2D class sequence feature extractor based on the CNN algorithm, 13 general features from fourteen healthy human subjects and two patients (one with myasthenia gravis and the other with the laryngeal cancer) can be classified with a high accuracy of 98.2%.”

“In the proof-of-the-concept demonstration, five Chinese pinyin and five vowels are chosen as the acoustic states, together with swallowing, drinking water, and coughing behaviors as feature states from fourteen healthy human subjects and two patients (one with myasthenia gravis and the other with the laryngeal cancer, sampling rate of 333 Hz).”

“The high-sensitivity accelerometer ADXL-345 with a sampling frequency of up to 800 Hz in the patch allows successful continuous monitoring of various activities, such as sitting, talking, swallowing, walking, and jumping (Fig. 4a and movie S4-6), with a wide frequency spectrum from 0 to 400 Hz (Fig. 4b). The simultaneously measured acceleration data along three different directions allow the integrated system to distinguish multiple motions separately (e.g., talking while walking, drinking water while coughing, drinking water while swallowing) (Figs. S21 and S22), as well as capture the swallowing process from a patient with myasthenia gravis (Fig. S23).”

Comment#6. Is there enough detail provided in the methods for the work to be reproduced? no really, as the limitation of data sizes, detailed evaluation of using the proposed sensors, should be further elaborated.

Our response: We thank the reviewer for this comment. We have carried out testing of the device with another 11 human subjects (including two patients who suffer from the myasthenia gravis and the laryngeal cancer, respectively). The swallowing signals over several repetitions from a patient with myasthenia gravis and 9 health adults are captured and compared (Fig. S23),

indicating higher muscle force (acceleration along z-axis) to complete swallowing in the patient with myasthenia gravis. Although the captured data can help build the individualized database of the laryngeal postoperative signature, the rehabilitation is out of scope of this work and it will be pursued in our future studies.

In addition, the high reproducibility of the fabrication method is highlighted by the batch production of the integrated device platform (Fig. S37).

Fig. S23. Normalized swallowing signals of 9 healthy subjects and the patient with myasthenia gravis when they swallow with a repetition.

Fig. S37. Optical image of the integrated device platform from batch production.

Our revision to the manuscript:

“Finally, spin-coating silicone elastomer on the bare circuit at 500 rpm for 10 s and curing at 40°C for 1.5 hrs completed the fabrication. The batch production of the integrated device platform highlighted the high reproducibility of the fabrication method (Fig. S37).”

Reviewer #3 (Remarks to the Author):

Thank you for the opportunity to review the paper titled “A fully integrated, standalone stretchable device platform with in-person adaptive machine learning for rehabilitation.” This manuscript reports the design and validation process of a new wireless sensor. This reviewer applauds the authors on their work in this important area. However, this reviewer’s enthusiasm for the work is lowered by a few concerns.

Our response: We highly appreciate the reviewer’s positive evaluation of our work. We also appreciate the insightful comments that help significantly improve the overall quality of this work.

Comment #1. *The authors would benefit from working with a swallowing specialist as there are some areas that throughout the manuscript that requires input from a swallowing specialist. For example, on page 3, line 49, the authors use the words “genetic choking.” It is not clear what genetic choking means. Furthermore, swallowing disorders can manifest as a result of many disorders including neurological disorders. It is not only related to “neck cancer or post-surgical treatment.” Similarly, the title of the section on page 8, line 193 is post-surgical rehabilitation. However, post-surgical patients represent only part of the patients who receive swallowing rehabilitation. Furthermore, authors should also rephrase the sentence in lines 194-196 as it is not accurate. In addition, the difference between drinking water and swallowing is not clear to this reviewer.*

Our response: We thank the reviewer for the suggestion. We have replaced “genetic choking” with congenital choking after checking with two swallowing specialists.

The difference between drinking water and swallowing lies in the duration and signal patterns (**Fig. S22**). The process of drinking water consists of holding water in the mouth lasting for 840 ms, flowing along the throat for 150 ms with a large acceleration magnitude, and swallowing through the pear-shaped postcricoid area for 270 ms (**Fig. S22a**). In contrast, swallowing only exhibits

swallowing through the pear-shaped postcricoid area for 490 ms with a relatively smaller acceleration (**Fig. S22b**).

Fig. S22. Comparison of the acceleration responses between a drinking water and b swallowing.

Our revision to the manuscript: We have included the new results in the revised manuscript.

“For those who suffer from congenital choking^{5,6} or neck cancer⁷⁻⁹, post-surgical rehabilitation of the human throat often requires the clinician’s continuous monitoring and evaluation of swallowing ability¹⁰, vocal-fold motion¹¹, oral intake of liquid¹², and others¹³⁻¹⁵.”

“The machine learning model for rehabilitative evaluation

The clinical evaluation for laryngeal rehabilitation usually focuses on typical events (i.e., talking, swallowing, volume and viscosity swallowing with modified safety and effectiveness indicators) with long-term and extensive efforts⁷⁷.”

“The high-sensitivity accelerometer ADXL-345 with a sampling frequency of up to 800 Hz in the patch allows successful continuous monitoring of various activities, such as sitting, talking, swallowing, walking, and jumping (**Fig. 4a** and **movie S4-6**), with a wide frequency spectrum from 0 to 400 Hz (**Fig. 4b**). The simultaneously measured acceleration data along three different

directions allow the integrated system to distinguish multiple motions separately (e.g., talking while walking, drinking water while coughing, drinking water while swallowing) (Figs. S21 and S22), as well as capture the swallowing process from a patient with myasthenia gravis (Fig. S23).”

Comment #2. Page 2, Line 33: The suprasternal notch refers to a specific anatomic location (the notch above the sternum). The location that the sensor was placed is not the suprasternal notch area.

Our response: We thank the reviewer for this comment. We have revised the term from “suprasternal throat” to “throat” in the revised manuscript.

Our revision to the manuscript:

“Here, we report the design and validation of a fully integrated standalone stretchable device platform that provides wireless measurements and machine learning-based analysis of diverse vibrations and muscle electrical activities from the throat.”

Comment#3. Page 3, lines 66-68: Authors refer to flexible EMGs that were developed (22, 24), however, mention that these suffer from low signal quality, motion artifacts, poor system integration, and severe skin inflammation. This reviewer recommends reviewing the citations 22 and 24 and reporting the findings accurately.

Our response: We thank the reviewer for the careful check. We have carefully reviewed these two references (ref #22 is now #23) and updated the discussion in the revised manuscript.

Our revision to the manuscript: We have carefully reviewed these two references and updated the discussion in the revised manuscript.

Efforts to address this challenge lead to the development of a wearable accelerometer for neck voice disorders²⁴ and flexible surface electromyogram (sEMG) electrodes with a strain sensor for oropharyngeal swallowing disorders^{23, 25}. However, these flexible sensors suffer from limited **stretchability of less than 16%**^{23, 25}, poor system integration **with only sensors without functional circuit board**^{21, 26}, and severe skin inflammation or allergy **during use over several hours** owing to its low permeability²⁷⁻²⁹.

***Comment #4.** The main concern about this paper is in the rationale (71-76). The authors report that there are skin-mounted mechano-acoustic sensors that track activity at the suprasternal notch and that high-quality sEMG signals have not been integrated into these platforms. While this is true, it is important to note that the suprasternal notch area or the laryngeal location that the authors mounted the device are not ideal locations to monitor swallowing activity. It has been well established that larynx moves throughout the swallow including before and after the swallow because of motion unrelated to swallowing. For example, chewing results in the inferior and superior movements of the larynx that are similar to the movements observed during swallowing. Furthermore, it has been well established through intramuscular electrodes, surface electrodes and imaging technologies that the ideal location for capturing swallowing movements using sEMG is the suprahyoid area. This reviewer is concerned about the location of the sensor and the integrity of the signals that were obtained using the sensors. I recommend the authors to use imaging technologies such as VFSS (the current gold-standard method) to validate their signals and ensure that the signals that they are reporting are actually swallowing signals.*

Our response: We highly appreciate for the reviewer's comments and suggestions. The swallowing signature can be clearly observed by placing the integrated device at P2 with comparable performance to the suggested location of P3 (the suprahyoid area), whereas the signal at P1 (suprasternal notch area) is almost inconspicuous (**Fig. S26**), leading to the choice of P1 for

swallowing detection. Furthermore, considering that P3 is not suitable for respiration measurement, P2 is chosen in this study (**Fig. S25**).

Although the VFSS is a gold-standard method, it is associated with high-X ray radiation during the long-time test (over 30 mins), posing health concerns. As an alternative, the fiberoptic endoscopic examination of swallowing (FEES) serves as another safe and comfortable gold-standard method, which is used to investigate and validate our device for swallowing studies (**Fig. R8**).

Fig. S26. Three marked positions on a healthy human male for swallowing tests (left) and the signal measured by our integrated device platform (right).

The swallowing process with the pear-shaped postcricoid area transitioned from opening to closing can be accurately monitored by our integrated wireless platform, which is also validated against the gold standard based on the fiberoptic endoscopic examination of swallowing (FEES). Compared with the healthy control (**Fig. S34a**), the patient with myasthenia gravis shows higher muscle force to complete swallowing (**Fig. S34b**, green dashed ellipse). In comparison, the FEES is not able to capture this difference.

Fig. S34. Comparison between a healthy control and b patient with myasthenia gravis during swallowing tests captured by our integrated wireless platform and the FEES: Real-time screenshot on the App interface (left), image showing the device at the oropharyngeal junction area and the FEES into the esophagus (middle), and FEES images of the closing process of the pear-shaped postcricoid area (right).

The volume and viscosity swallowing test-chinses version (VVST-CV) with modified safety and effectiveness indicators is used to quantify the analysis as the subject ingests a blue edible indicator with a body weight percentage of 2% (stage #1) and 1% (stage #2) (**Fig. S35a**). The clinical standard for swallowing rhythm requires swallowing to be less than 3 s from the the indicator in mouth to the circumpharyngeal muscle opening and a total time of less than 15 s for the complete ingestion process. The swallowing in both stages for the patient with myasthenia gravis exceeds 15 s (**Fig. S35b**). Liquid residual still exists after 25 s in the oropharyngeal junction area in stage #1 (**Fig. S35c**). The duration of the process of about 10 s captured by our device is consistent with the clinical standard (**Fig. S35d**), demonstrating the feasibility and reliability of the integrated system. It is important to note that one portion of the ingesting process (7-12s) in the stage #2 is not captured by the FEES due to the closed oropharyngeal junction area to block the endoscopic view (**Fig. S35e**). In contrast, this missing swallowing process (green dashed ellipse) is still clearly recorded by our device (**Fig. S35f**), providing enhanced diagnostic accuracy for laryngeal

postoperative patients.

Fig. S35. Volume and viscosity swallowing test with modified safety and effectiveness indicators (VVST-CV) for the patient with myasthenia gravis. a Optical images showing the patient swallowing a blue edible indicator with a weight rate of 2% (left, stage #1) and 1% (right, stage #2). **b** Real-time signals captured by our device during the two-stage swallowing. **c** FEES images in stage #1 and **d** the corresponding time-interval as a function of the number of swallowing. **e** FEES images in stage #2 and **f** the time-interval as a function of number of swallowing.

Our revision to the manuscript: We have included the justification of the chosen measurement location for the integrated device in the revised manuscript.

“As the mounting position of the patch moves up the laryngeal skin, the measured cardiac behavior

remains unchanged, but the respiration amplitude decreases (**Fig. S25**) due to significantly reduced movements (along the neck direction) farther away from the chest cavity. **Moreover, the swallowing signature can be clearly observed by placing the integrated device at the throat area with comparable performance to the suprahyoid area, whereas the signal at the suprasternal notch area is almost inconspicuous (Fig. S26), leading to the choice of middle throat area for laryngeal detection.**”

Comment #5. The authors report data from 2 healthy adults. However, they also make claims such as “to help evaluate the post-surgical state...” page 10, line 233. It is not clear to this reviewer how authors are able to make these generalizations by using the data obtained from healthy individuals who did not receive head and neck surgery.

Our response: We thank the reviewer for this comment. We have updated this statement (with tested data from fourteen health subjects) in the revised manuscript to help avoid confusion. Furthermore, we have carried out testing of the device with another 11 human subjects (including two patients who suffer from the myasthenia gravis and the laryngeal cancer, respectively).

The swallowing process with the pear-shaped postcricoid area transitioned from opening to closing can be accurately monitored by our integrated wireless platform, which is also validated against the gold standard based on the fiberoptic endoscopic examination of swallowing (FEES). Compared with the healthy control (**Fig. S34a**), the patient with myasthenia gravis shows higher muscle force to complete swallowing (**Fig. S34b**, green dashed ellipse). In comparison, the FEES is not able to capture this difference.

Fig. S34. Comparison between a healthy control and b patient with myasthenia gravis during swallowing tests captured by our integrated wireless platform and the FEES: Real-time screenshot on the App interface (left), image showing the device at the oropharyngeal junction area and the FEES into the esophagus (middle), and FEES images of the closing process of the pear-shaped postcricoid area (right).

The volume and viscosity swallowing test-chinses version (VVST-CV) with modified safety and effectiveness indicators is used to quantify the analysis as the subject ingests a blue edible indicator with a body weight percentage of 2% (stage #1) and 1% (stage #2) (**Fig. S35a**). The clinical standard for swallowing rhythm requires swallowing to be less than 3 s from the the indicator in mouth to the circumpharyngeal muscle opening and a total time of less than 15 s for the complete ingestion process. The swallowing in both stages for the patient with myasthenia gravis exceeds 15 s (**Fig. S35b**). Liquid residual still exists after 25 s in the oropharyngeal junction area in stage #1 (**Fig. S35c**). The duration of the process of about 10 s captured by our device is consistent with the clinical standard (**Fig. S35d**), demonstrating the feasibility and reliability of the integrated system. It is important to note that one portion of the ingesting process (7-12s) in the stage #2 is not captured by the FEES due to the closed oropharyngeal junction area to block the endoscopic view (**Fig. S35e**). In contrast, this missing swallowing process (green dashed ellipse) is still clearly recorded by our device (**Fig. S35f**), providing enhanced diagnostic accuracy for laryngeal

postoperative patients.

Fig. S35. Volume and viscosity swallowing test with modified safety and effectiveness indicators (VVST-CV) for the patient with myasthenia gravis. a Optical images showing the patient swallowing a blue edible indicator with a weight rate of 2% (left, stage #1) and 1% (right, stage #2). **b** Real-time signals captured by our device during the two-stage swallowing. **c** FEES images in stage #1 and **d** the corresponding time-interval as a function of the number of swallowing. **e** FEES images in stage #2 and **f** the time-interval as a function of number of swallowing.

Furthermore, the device is also applied to record the swallowing process for an old patient with a recent operation for the laryngeal cancer (Fig. R4). The leaked liquid (to the trachea) and residual at the oropharyngeal junction area result in an abnormal swallowing process (incomplete

swallowing of the blue indicator due to with severe coughing) with longer duration for each swallowing/ingestion and severe coughing.

Fig. R4. VVST-CV for an old patient with a clinical operation for laryngeal cancer. Optical image of the patient swallowing a blue edible indicator (left), real-time data displayed on the APP (middle), and FEES images during swallowing (right).

In addition, the swallowing signals over several repetitions from a patient with myasthenia gravis and 9 healthy subjects are captured and compared (Fig. S23), indicating higher muscle force (acceleration along z-axis) to complete swallowing in the patient with myasthenia gravis. Although the captured data can help build the individualized database of the laryngeal postoperative signature, the rehabilitation is out of scope of this work and it will be pursued in our future studies.

Fig. S23. Normalized swallowing signals of 9 healthy subjects and the patient with myasthenia gravis when they swallows with an repetition.

Our revision to the manuscript: We have updated the statement and included the new results in the revised manuscript.

“To help automatically evaluate the laryngeal condition from the new patients and healthy individuals, a CNN-based 2D-like sequential feature extractor (2D-SFE) is explored to classify and infer pathological status based on the classification of physiological events (**Fig. 5a**).

“The high-sensitivity accelerometer ADXL-345 with a sampling frequency of up to 800 Hz in the patch allows successful continuous monitoring of various activities, such as sitting, talking, swallowing, walking, and jumping (**Fig. 4a** and **movie S4-6**), with a wide frequency spectrum from 0 to 400 Hz (**Fig. 4b**). The simultaneously measured acceleration data along three different directions allow the integrated system to distinguish multiple motions separately (e.g., talking while walking, drinking water while coughing, drinking water while swallowing) (**Figs. S21** and **S22**), as well as capture the swallowing process from a patient with myasthenia gravis (**Fig. S23**).”

Comment #6. Page 15, lines 356-361: The authors report that the pathological degree was sorted into eight levels. Are the authors referring to severity? If so, it would be important to clarify how these 8 levels were determined for swallowing, drinking water, and talking. This reviewer is not familiar with any 8-level evidence-based severity ratings related to swallowing and speech impairments.

Our response: Many thanks for the comment. During laryngeal rehabilitation, various behaviors such as swallowing, talking, and drinking water are assessed to be either normal or abnormal for each behavior, so the combined evaluation results of the three representative behaviors form eight

evaluation states/degrees (**Table S4**). Although this 8-level/degree rating is not currently used in the clinical evaluation, we hope that it could provide insights to infer the patient’s conditions and guide individualized rehabilitation in the future.

Table S4 Pathological degree evaluation according to the health condition of three typical events below			
Behavior Degree	Swallowing ability (S)	Drinking water ability (D)	Talking ability (T)
I	√	√	√
II	√	√	×
III	√	×	√
IV	√	×	×
V	×	√	√
VI	×	√	×
VII	×	×	√
VIII	×	×	×

× Abnormal, √ Normal, Meaning that the condition of the corresponding ability is diagnosed by laryngeal rehabilitative standard during clinical assessments.

Our revision to the manuscript: We have discussed the possible use of the 8-degree evaluation for future individualized rehabilitation in the revised manuscript.

“The pathological degree at the cloud server interface was sorted into eight levels (from the best state I to the worst state VIII) according to the rehabilitative conditions of three processes:

swallowing (S), drinking water (D), and talking (T). During laryngeal rehabilitation, various behaviors such as swallowing, talking, and drinking water are assessed to be either normal or abnormal for each behavior, so the combined evaluation results of the three representative behaviors form eight evaluation states (**Table S4**). Although this 8-degree rating is not currently used in the clinical evaluation, it could provide insights of patient's conditions to help guide individualized rehabilitation in the future.”

REVIEWER COMMENTS

Reviewer #1 (Remarks to the Author):

All comments from the reviewer were addressed well in the revised manuscript which is now ready for publication.

Reviewer #2 (Remarks to the Author):

thanks authors has addressed most of my previous comments, regarding the power consumption question, I would like to further clarify few points, authors mentioned the device could be powered by battery for 5-6 hours, but there is lack of evidence or performance metrics to show this is fixable, for example, measurement of power consumption or current recording etc. this is important to show how this device is in advance or could be used for future comparison with other similar device in terms of performance.

Reviewer #1 (Remarks to the Author):

All comments from the reviewer were addressed well in the revised manuscript which is now ready for publication.

Our response: We highly appreciate the reviewer's previously positive evaluation of our work. We also appreciate these insightful comments that truly help significantly improve the overall quality of this work.

Reviewer #2 (Remarks to the Author):

Thanks authors has addressed most of my previous comments, regarding the power consumption question, I would like to further clarify few points, authors mentioned the device could be powered by battery for 5-6 hours, but there is lack of evidence or performance metrics to show this is fixable, for example, measurement of power consumption or current recording etc. this is important to show how this device is in advance or could be used for future comparison with other similar device in terms of performance.

Our response: We appreciate the referee's positive evaluation and the valuable comments that have enhanced the quality of our work. Our integrated device can operate in two modes: a high-power connection mode and a low-power sleeping mode, enabled by the Bluetooth Low Energy (BLE) communication circuit, as shown in **Fig. R1**. The connection interval (t_1) of PW02 module is about 7.5~15 ms as the standard BLE protocol stack (reference as https://softwaredl.ti.com/lprf/simplelink_cc2640r2_sdk/1.00.00.22/exports/docs/blestack/html/ble-stack/index.html) and the sleeping interval (t_2) is about three times more than the connection interval ($t_1 : t_2 \approx 1:3$, this can be set in the software according to the transmission speed of the wireless signal). The BLE host regularly send connection events to smart phone or sub-hub (CP2102-HM-10, BLE4.0). In the connection and sleeping modes, the power consumption is 19.6 mW (voltage 2.8 V and current 7 mA, **Fig. R2a**) and 2.71 mW (voltage 2.71 V and current 1 mA, **Fig. R2b**), respectively. The indicator LED in **Fig. R2**'s bottom enlarged images confirms these modes. With a Li⁺ battery of 35 mA h capacity, the device's working time (T) can be calculated as:

$$T = T_1 + T_2 = 4T_1$$

where the T_1 is connection time, T_2 is sleeping time, T is the operation time of the device. According to the ideal working time of the device operating at connection model,

$$T_1 = \frac{35 \text{ mA h}}{19.6 \text{ mW}} \approx 1.79 \text{ h}$$

Hence,

$$T = 4T_1 = 7.14 \text{ h}$$

However, practical factors such as wireless scanning and reconnection affect power consumption, making $T < 7.14 \text{ h}$. Based on the practical power consumption, the working time can be calculated as:

$$19.6 \text{ mW} \times T_1 + 2.71 \text{ mW} \times 3T_1 \approx 35 \text{ mW h}$$

where $T_1 \approx 1.26 \text{ h}$, so working time $T \approx 5.04 \text{ h}$ belongs to the range of 5~6 h. The calculated time highly agrees with the tested result. It's important to note that various factors, including the device's proximity to the wireless terminal receiver, the embedded main program, and the communication type (I2C, SPI, and UART, etc.), influence power consumption. To extend working time, wireless battery charging presents a promising solution [Xu, S., Zhang, Y., Cho, J. et al. Nat Commun 4, 1543 (2013)].

Fig. R1. Current draw of the PW02 module in a low-power consumption state within the standard BLE protocol stack.

Fig. R2. Optical images of the integrated device operating in a high-power connection and b low-power connection mode (the enlarged images are their corresponding working indicator's states.)

Our revision to the manuscript: We have included the working time of the device in the Methods section of the revised manuscript.

“The Bluetooth module PW02 (Phangwei Link) had serial port transmission ability. **The integrated device can maintain continuous operation for approximately 5 to 6 hours on a 35 mA h Li battery.**”

REVIEWERS' COMMENTS

Reviewer #2 (Remarks to the Author):

thanks for the detailed answers, I am happy with the current version.